# Repurposing High-Throughput Screening Identifies Unconventional Drugs with Antibacterial and Antibiofilm Activities against *Pseudomonas aeruginosa* under Experimental Conditions Relevant to Cystic Fibrosis

Giovanni Di Bonaventura,[a,b] Veronica Lupetti,[a,b] Andrea Di Giulio,[c] Maurizio Muzzi,[c] Alessandra Piccirilli,[d] Lisa Cariani,[e] Arianna Pompilio[a,b]

aDepartment of Medical, Oral, and Biotechnological Sciences, G. d'Annunzio University of Chieti-Pescara, Chieti, Italy
bCenter for Advanced Studies and Technology, G. d'Annunzio University of Chieti-Pescara, Chieti, Italy
cDepartment of Science, University Roma Tre, Rome, Italy
dDepartment of Biotechnological and Applied Clinical Sciences, University of L'Aquila, L'Aquila, Italy
eMicrobiology Unit, Fondazione IRCCS Ca' Granda Ospedale Maggiore Policlinico, Milan, Italy

**ABSTRACT** *Pseudomonas aeruginosa* is the most common pathogen infecting cystic fibrosis (CF) lungs, causing acute and chronic infections. Intrinsic and acquired antibiotic resistance allow *P. aeruginosa* to colonize and persist despite antibiotic treatment, making new therapeutic approaches necessary. Combining high-throughput screening and drug repurposing is an effective way to develop new therapeutic uses for drugs. This study screened a drug library of 3,386 drugs, mostly FDA approved, to identify antimicrobials against *P. aeruginosa* under physicochemical conditions relevant to CF-infected lungs. Based on the antibacterial activity, assessed spectrophotometrically against the prototype RP73 strain and 10 other CF virulent strains, and the toxic potential evaluated toward CF IB3-1 bronchial epithelial cells, five potential hits were selected for further analysis: the anti-inflammatory and antioxidant ebselen, the anticancer drugs tirapazamine, carmofur, and 5-fluorouracil, and the antifungal tavaborole. A time-kill assay showed that ebselen has the potential to cause rapid and dose-dependent bactericidal activity. The antibiofilm activity was evaluated by viable cell count and crystal violet assays, revealing carmofur and 5-fluorouracil as the most active drugs in preventing biofilm formation regardless of the concentration. In contrast, tirapazamine and tavaborole were the only drugs actively dispersing preformed biofilms. Tavaborole was the most active drug against CF pathogens other than *P. aeruginosa*, especially against *Burkholderia cepacia* and *Acinetobacter baumannii*, while carmofur, ebselen, and tirapazamine were particularly active against *Staphylococcus aureus* and *B. cepacia*. Electron microscopy and propidium iodide uptake assay revealed that ebselen, carmofur, and tirapazamine significantly damage cell membranes, with leakage and cytoplasm loss, by increasing membrane permeability.

**IMPORTANCE** Antibiotic resistance makes it urgent to design new strategies for treating pulmonary infections in CF patients. The repurposing approach accelerates drug discovery and development, as the drugs' general pharmacological, pharmacokinetic, and toxicological properties are already well known. In the present study, for the first time, a high-throughput compound library screening was performed under experimental conditions relevant to CF-infected lungs. Among 3,386 drugs screened, the clinically used drugs from outside infection treatment ebselen, tirapazamine, carmofur, 5-fluorouracil, and tavaborole showed, although to different extents, anti-*P. aeruginosa* activity against planktonic and biofilm cells and broad-spectrum activity against other CF pathogens at concentrations not toxic to bronchial epithelial cells. The mode-of-action studies revealed

Address correspondence to Giovanni Di Bonaventura, gdibonaventura@unich.it.

The authors declare no conflict of interest.

ebselen, carmofur, and tirapazamine targeted the cell membrane, increasing its permeability with subsequent cell lysis. These drugs are strong candidates for repurposing for treating CF lung *P. aeruginosa* infections.

**KEYWORDS** *Pseudomonas aeruginosa*, cystic fibrosis, antibiotic resistance, drug repurposing, antibacterial, antibiofilm, biofilms

*P*seudomonas aeruginosa is the most prevalent respiratory pathogen in adult cystic fibrosis (CF) patients. It is the leading cause of morbidity and mortality due to a progressive pulmonary decline secondary to frequent acute pulmonary exacerbations (1). Once the infection is fully established, *P. aeruginosa* eradication cannot be achieved. Therefore, in CF patients, antimicrobial therapy aims to reduce the bacterial density in the respiratory tract (2). Despite the demonstrated efficacy of regimens based on cycles of nebulized tobramycin in reducing bacterial loads, repeated exposure selects many resistant isolates (3). In addition, the attitude of *P. aeruginosa* to form intrinsically antibiotic-resistant biofilms during chronic infection makes treatment of exacerbations more challenging (1). This scenario is further complicated by evidence that at the site of infection, bacteria grow under conditions (e.g., highly viscous sputum, acidic pH, anaerobiosis) affecting both the delivery and functionality of antibiotics (4, 5). As such, adequate treatment options are limited and new compounds with potent anti-*P. aeruginosa* activity are needed urgently.

Drug discovery is a crucial process to discover compounds and molecules that may develop into clinical therapeutic drugs; however, it is expensive and full of risks of failure (6). In this framework, the redirection or repositioning of nonantibiotic drugs is a promising alternative since reusing drugs promotes the accelerated exploration of new properties, making the process less expensive than the traditional discovery of new drugs with antimicrobial effects (7). In addition, repositioning is more likely to produce bioavailable and safe compounds, as their toxicity has been known and reported for years (7). Several drugs belonging to the classes of antidepressants, antineoplastics, antacids, and hypoglycemic agents showed varied antimicrobial effects, and some against multidrug-resistant (MDR) microorganisms, thus confirming the repositioning of existing drugs for antimicrobial purposes (8).

In this work, we used a high-throughput screening (HTS) assay to screen over 3,300 compounds, mostly already launched, to find drugs with novel antimicrobial activity against *P. aeruginosa*. After evaluating their antibacterial effect against the prototype RP73 strain and 10 current CF strains, along with the toxic potential toward a cell type relevant to CF, five potential hits were selected for further analysis by the following methods: (i) bactericidal activity through a time-kill assay; (ii) synergism with tobramycin using a checkerboard assay, (iii) antibiofilm activity, using both viable cell count and spectrophotometric assays; and (iv) activity against pathogenic CF species other than *P. aeruginosa*. Finally, the mechanism of action underlying the anti-*P. aeruginosa* effect was evaluated by electron microscopy and quantitative methods to assess cell membrane damage and reactive oxygen species (ROS) production. All assays were carried out for the first time under physicochemical conditions relevant to CF lung to give higher therapeutic predictive value to our findings. We selected tobramycin as a model antibiotic since it is the first-line therapy against *P. aeruginosa* lung infections (9).

## RESULTS AND DISCUSSION

**HTS screening under "CF-like" conditions.** In the present study, we screened under "CF-like" experimental conditions the Drug Repurposing Compound Library (MedChem Express), consisting of 3,386 bioactive compounds (with 2,342 already launched and 1,044 that have reached clinical trial stages in the United States: 1 drug in phase I, 606 drugs in phase II, 372 drugs in phase III, and 65 drugs in phase IV) used as drugs with several therapeutic indications, including cancer, neurodegenerative, infectious, and cardiovascular diseases.

To our knowledge, this is the first study in which an HTS is performed under experimental conditions simulating the actual physicochemical characteristics of the CF lung ecosystem that could affect antibiotic effectiveness at the site of infection (4, 5). First, the efficacy of antimicrobial agents should be evaluated in a chemically defined medium mimicking

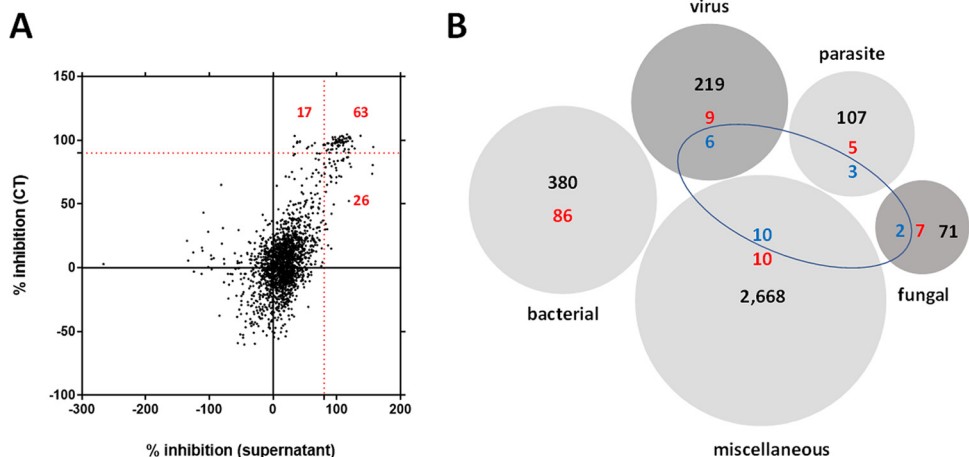

**FIG 1** Primary screening and breakdown of the Drug Repurposing Compound Library (MedChem Express) by category. (A) A total of 106 hits (red highlighted) were found causing a *P. aeruginosa* RP73 growth inhibition higher than 90% (dotted red lines). Each compound was tested for antibacterial activity at 0.1 mM, both on supernatant and by a tetrazolium-based CellTiter (CT) assay. Results are expressed as the mean percentage of inhibition versus unexposed control. (B) The library consisted of 3,386 compounds with the following primary targets: bacterial, 380; virus, 219; parasite, 107; fungal, 71; and miscellaneous, 2,668. The preliminary screening revealed 106 hits (highlighted in red), 14 of which (highlighted in blue) were identified as repositionable drug candidates. Some hits had multiple targets.

the nutritional composition of CF sputum. Several formulations of artificial sputum medium (ASM) were developed over the years using different combinations of mucin, extracellular DNA, iron, amino acids, and lipids (10). Second, the infected endobronchial mucus of CF patients contains hypoxic/anaerobic zones mainly generated by the respiratory burst of polymorphonuclear leukocytes and the $O_2$ consumption by respiratory epithelium (11). Third, it has been observed that the epithelial lining fluid of patients with CF is acidified, partly due to inflammation (12). The average pH of submucosal gland fluid in CF patients is ~6.6 to 7.0 (10). Both atmosphere and pH values were recognized among the most critical factors known to affect the activity of antibiotics (4, 13, 14).

Therefore, to accurately judge the feasibility of the hits as potential anti-infectives in the context of CF, we investigated their activity in a medium proposed by Sriramulu et al. (15), recognized as being a suitable model of chronic lung colonization useful for evaluating therapeutic procedures and studying antibiotic resistance mechanisms (10), under acidic pH (6.8) and reduced $O_2$ tension (5% $CO_2$).

**Identification of potential anti-*P. aeruginosa* hits.** We initially identified compounds active against the *P. aeruginosa* RP73 strain through a 96-well microtiter plate-based HTS of 3,386 compounds belonging to the compound library. All compounds were initially tested at a single concentration of 0.1 mM to pinpoint active hits. The results were expressed as the percentage of inhibition of bacterial growth compared to that in untreated controls, estimated from spectrophotometric readings carried out on supernatant and using the CellTiter (CT) assay (Fig. 1).

Based on this primary screen—carried out in triplicate in a single independent experiment—we identified 2,641 compounds able to affect *P. aeruginosa* RP73 growth, although to different extents: 1,783 with low activity (i.e., growth reduction of <25%), 640 with moderate activity (i.e., growth reduction of ≥25% and <60%), 112 with high activity (i.e., growth reduction of ≥60% and <90%), and, more interestingly, 106 compounds with excellent activity (i.e., growth reduction of ≥90%) (Fig. 1A).

The complete list of the 106 compounds showing excellent activity is provided in Table S1 in the supplemental material. Based on their known pharmacological profiles, these compounds were grouped into different classes based on the primary target, considering that some compounds had multiple targets (Fig. 1B): 86 were found among a total of 380 (22.6%) drugs with a "bacterial" primary target, validating the high fidelity of the antibacterial screening; 9 out of 219 (4.1%) drugs with a "virus" primary target; 5 out of 107 (4.6%) drugs

with a "parasite" target; and 7 out of 71 (9.8%) drugs with "fungal" primary target. However, perhaps strictly from the drug repurposing point of view, 10 compounds were found among the 2,268 (0.4%) multifunctional drugs used for other applications but with no known or characterized antimicrobial activity to date.

Because this work aimed to find new repositionable drugs, we decided to move forward only with those having a primary target different from "bacterial." Twenty-three compounds meeting this criterion were initially selected after the primary screen and retested in a secondary screen—carried out in two independent experiments, each in triplicate, confirming 14 compounds. These repositionable hit candidates comprise six groups based on their known function and clinical indications (Table 1). They include the following: (i) six anticancer drugs (triapine [3-AP], tirapazamine, RRx-001, panobinostat, carmofur, and 5-fluorouracil); (ii) sulforaphane, resveratrol, and ebselen, with antioxidant, anti-inflammatory, and anticancer properties; (iii) the antirheumatic auranofin; (iv) the antioxidant amino acid L-selenomethionine; (v) two antiparasitic drugs (bithionol and broxyquinoline); and (vi) the antifungal tavaborole. All were launched except for 3-AP, ebselen, tirapazamine, RRx-001, and sulforaphane, which are currently in phase III clinical trials. At first glance, the active compounds seem to be highly scattered structurally. However, the most active molecules can behave as electrophiles, undergoing the attack of nucleophilic groups (typically cysteine).

The hit rate we observed is surprisingly high (0.4% [14 out of 3,386]), probably because the library we tested consists mainly of approved drugs. Typical hit rates of <0.1% were reported for HTS of large, random chemical libraries of small synthetic molecules that commonly contain numerous non-drug-like molecules (16, 17).

The screening quality was evaluated using the $Z$-factor, a standard measure of the robustness and feasibility of an HTS, estimating the magnitude of the difference between the positive and negative controls relative to the sum of the respective standard deviations (SDs). The average $Z$-factor between the negative and positive controls in the 96-well test plates was 0.74 (range, 0.55 to 0.92), well above the 0.50 value, indicating the assay could reliably separate positive and negative controls. In addition to the $Z$-factor, the coefficient of variation (<10%) and the signal-to-background ratio (>10-fold) further supported the feasibility of our anti-*P. aeruginosa* drug screening assay.

**Impact of the experimental conditions on the hits' activity.** To evaluate the influence of culture conditions on the screening results, the minimal inhibitory concentration (MIC) values of the 14 hits active against *P. aeruginosa* RP73 were also measured under "standard" conditions—i.e., those recommended by CLSI for susceptibility testing—using cation-adjusted Mueller-Hinton broth, neutral pH, and aerobic atmosphere.

A comparative evaluation showed that MICs are different (≥4-fold, ≥2 doubling dilutions) in most cases (12 out of 14 hits [85.7%]), with values consistently higher under "standard" conditions (range, 4 to ≥8-fold, 2 to ≥3 doubling dilutions) (Fig. 2). Conversely, MICs were similar (±1 doubling dilution) only in the cases of 3-AP and bithionol.

These findings demonstrated the importance of performing antibacterial assays under conditions simulating the host environment: in this case relevant to the CF lung since they possibly affect bacterial growth and antibiotic activity (4, 18–20).

In this regard, the present study may serve as a proof of concept for screening repurposing libraries under clinically relevant conditions, revealing that the activity of the hits against *P. aeruginosa* might be higher under conditions pertinent to the infected CF lung than those suggested for the conventional susceptibility testing methods. For example, the improved activity of tirapazamine under "CF-like" conditions might be due to its activation under hypoxic conditions (21). However, defining the respective roles of ASM components, acidic pH, and reduced $O_2$ concentration in the molecular mechanisms underlying the improved antibacterial activity was outside the present study's aim and warrants further investigations.

**Activity of potential hits against a set of *P. aeruginosa* strains from CF patients.** One of the hallmarks of *P. aeruginosa* is its highly dynamic genome. Specifically, the nutritionally restricted niche of the CF lung appears to be a key evolutionary force shaping the *P. aeruginosa* genome, as indicated by the genome plasticity of individual strains (22). This

**TABLE 1** Characteristics of the 14 primary hits selected for relevant activity (≥90%) against the *P. aeruginosa* RP73 strain

| Compound | Antibacterial activity (%)[a] | CAS no. | Mol wt | Target(s) | Pathway(s) | Chemical formula | Research area(s) | Clinical information |
|---|---|---|---|---|---|---|---|---|
| L-Selenomethionine | 95.1 | 3211-76-5 | 196.11 | Apoptosis, endogenous metabolite | Apoptosis, metabolic enzyme/protease | $C_5H_{11}NO_2Se$ | Cancer | Launched |
| Broxyquinoline | 100 | 521-74-4 | 302.95 | Parasite | Anti-infection | $C_9H_5Br_2NO$ | Infection | Launched |
| 3-AP | 97.5 | 143621-35-6 | 195.24 | DNA/RNA synthesis | Cell cycle/DNA damage | $C_7H_9N_5S$ | Cancer | Phase III |
| Ebselen | 100 | 60940-34-3 | 274.18 | Calcium channel, HIV, phosphatase, virus protease | Anti-infection, membrane transporter/ion channel, metabolic enzyme/protease, neuronal signaling | $C_{13}H_9NOSe$ | Cancer, infection, inflammation/immunology, neurological disease | Phase III |
| Tirapazamine | 100 | 27314-97-2 | 178.15 | Others | Others | $C_7H_6N_4O_2$ | Cancer | Phase III |
| RRx-001 | 100 | 925206-65-1 | 268.02 | Apoptosis, parasite | Anti-infection, apoptosis | $C_5H_6BrN_3O_5$ | Cancer, infection, immunology | Phase III |
| Tavaborole | 100 | 174671-46-6 | 151.93 | Antibiotic, fungal | Anti-infection | $C_7H_6BFO_2$ | Infection | Launched |
| Resveratrol | 100 | 501-36-0 | 228.24 | Antibiotic, apoptosis, autophagy, bacterial, fungal, IKK, Keap1-Nrf2, mitophagy, sirtuin | Anti-infection, apoptosis, autophagy, cell cycle/DNA damage, epigenetics, NF-κB | $C_{14}H_{12}O_3$ | Cancer, infection, inflammation/immunology | Launched |
| Panobinostat | 99.7 | 404950-80-7 | 349.43 | Apoptosis, autophagy, HDAC, HIV | Anti-infection, apoptosis, autophagy, cell cycle/DNA damage, epigenetics | $C_{21}H_{23}N_3O_2$ | Cancer | Launched |
| Carmofur | 100 | 61422-45-5 | 257.26 | Nucleoside antimetabolite/analog, SARS-CoV protease | Anti-infection, cell cycle/DNA damage | $C_{11}H_{16}FN_3O_3$ | Cancer | Launched |
| Auranofin | 100 | 34031-32-8 | 680.50 | Bacterial, SARS-CoV | Anti-infection | $C_{20}H_{34}AuO_9PS$ | Cancer, infection, inflammation/immunology | Launched |
| 5-Fluorouracil | 100 | 51-21-8 | 130.08 | Apoptosis, endogenous metabolite, HIV, nucleoside antimetabolite/analog | Anti-infection, apoptosis, cell cycle/DNA damage, metabolic enzyme/protease | $C_4H_3FN_2O_2$ | Cancer | Launched |
| Bithionol | 97.5 | 97-18-7 | 356.05 | Parasite | Anti-infection | $C_{12}H_6Cl_4O_2S$ | Cancer | Launched |
| Sulforaphane | 100 | 4478-93-1 | 177.29 | Apoptosis, HDAC, Keap1-Nrf2 | Apoptosis, cell cycle/DNA damage, epigenetics, NF-κB | $C_6H_{11}NOS_2$ | Cancer, inflammation/immunology | Phase III |

[a]The antibacterial activity, measured at 24 h post-exposure to each drug at 0.1 mM under "CF-like" conditions, is expressed as a percentage of the OD measured in the absence of drugs (100% in artificial sputum medium).

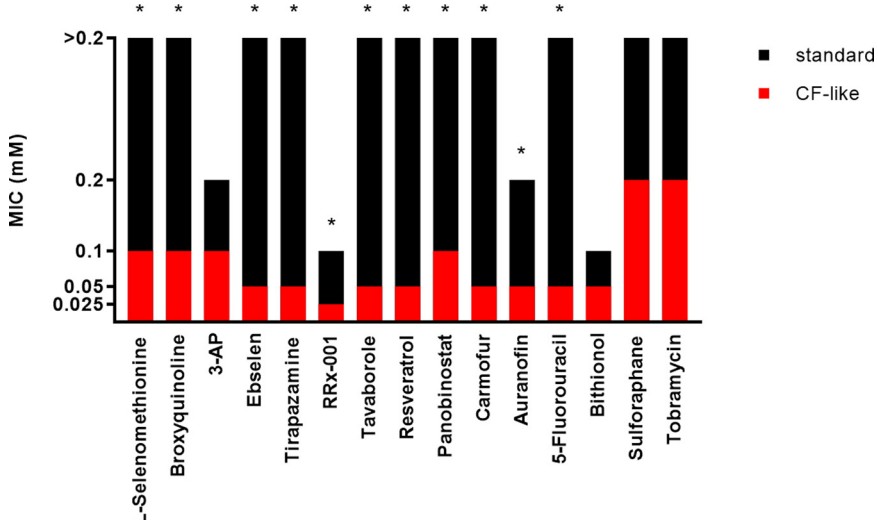

**FIG 2** Susceptibility of *P. aeruginosa* RP73 to the hits and tobramycin under different culture conditions. MIC values (millimolar concentrations) were measured by the broth microdilution technique under "standard" (cation-adjusted Mueller-Hinton broth [pH 7.2], aerobiosis) and "CF-like" (ASM [pH 6.8], 5% $CO_2$) experimental conditions. The asterisk indicates a significant difference in MIC values ($\geq$4-fold) between "standard" and "CF-like" settings.

heterogeneity among strains makes it necessary that a new potential antimicrobial be tested for its activity against a set of diverse and clinically relevant isolates. In this regard, the antibacterial activity of the 14 hit candidates was confirmed and evaluated in more detail by assessing MIC and minimal bactericidal concentration (MBC) values against the RP73 strain and 10 additional *P. aeruginosa* strains recently isolated from CF patients and chosen for their virulence and antibiotic resistance traits (Tables S2 and S3). The lowest concentrations of the compound capable of inhibiting the growth of 50% and 90% of the tested strains (MIC$_{50}$ and MIC$_{90}$, respectively) or at which no bacterial growth is evident (MBC$_{50}$ and MBC$_{90}$, respectively) were calculated to compare the activities of different drugs, and the results are summarized in Fig. 3 and 4.

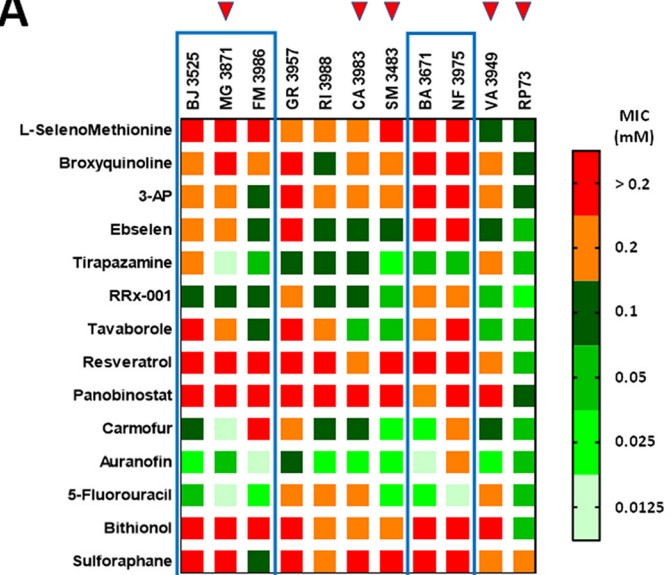

**B**

| Antibacterial hit | MIC range | MIC$_{50}$ | MIC$_{90}$ |
|---|---|---|---|
| L-SelenoMethionine | 0.1 - >0.2 (19.6 - >39.2) | 0.2 (39.2) | >0.2 (>39.2) |
| Broxyquinoline | 0.1 - >0.2 (30.3 - >60.6) | 0.2 (60.6) | >0.2 (>60.6) |
| 3-AP | 0.1 - >0.2 (19.5 - >39) | 0.2 (39.0) | >0.2 (>39.0) |
| Ebselen | 0.05 - >0.2 (13.7 - >54.8) | 0.1 (27.4) | >0.2 (>54.8) |
| Tirapazamine | 0.0125 - 0.2 (2.2 - 35.6) | 0.05 (8.9) | 0.2 (35.6) |
| RRx-001 | 0.025 - 0.2 (6.9 - 53.6) | 0.1 (26.8) | 0.2 (53.6) |
| Tavaborole | 0.05 - >0.2 (7.6 – 30.2) | 0.1 (15.6) | >0.2 (>30.2) |
| Resveratrol | 0.05 - >0.2 (11.4 - >45.6) | >0.2 (>45.6) | >0.2 (>45.6) |
| Panobinostat | 0.1 - >0.2 (34.9 - >69.8) | >0.2 (>69.8) | >0.2 (>69.8) |
| Carmofur | 0.0125 - 0.2 (3.2 - 51.4) | 0.1 (25.7) | 0.2 (51.4) |
| Auranofin | 0.0125 - 0.2 (8.4 - 135.6) | 0.025 (16.8) | 0.1 (67.8) |
| 5-Fluorouracil | 0.0125 - 0.2 (1.6 - 26) | 0.025 (3.2) | 0.2 (26.0) |
| Bithionol | 0.05 - >0.2 (17.8 - >71.2) | >0.2 (>71.2) | >0.2 (>71.2) |
| Sulforaphane | 0.1 - >0.2 (17.7 - >35.4) | >0.2 (>35.4) | >0.2 (>35.4) |
| Tobramycin | 0.025 - >0.2 (11,7 - >93.4) | 0.05 (23.8) | 0.1 (46.7) |

**FIG 3** *In vitro* activity of selected antibacterial hits against *P. aeruginosa* strains from CF patients. (A) Heat map of MIC data for the tested hit compounds. The values, expressed as millimolar concentrations, are color-coded, and color bars mark the matrix positions of compounds in a particular *P. aeruginosa* strain. Boxed are strains with strong pyocyanin, pyoverdine and protease production, while red triangles indicate MDR strains. (B) MIC range and the lowest concentrations that inhibited 50% and 90% of the tested strains (MIC$_{50}$ and MIC$_{90}$, respectively), expressed as millimolar concentrations (micrograms per milliliter).

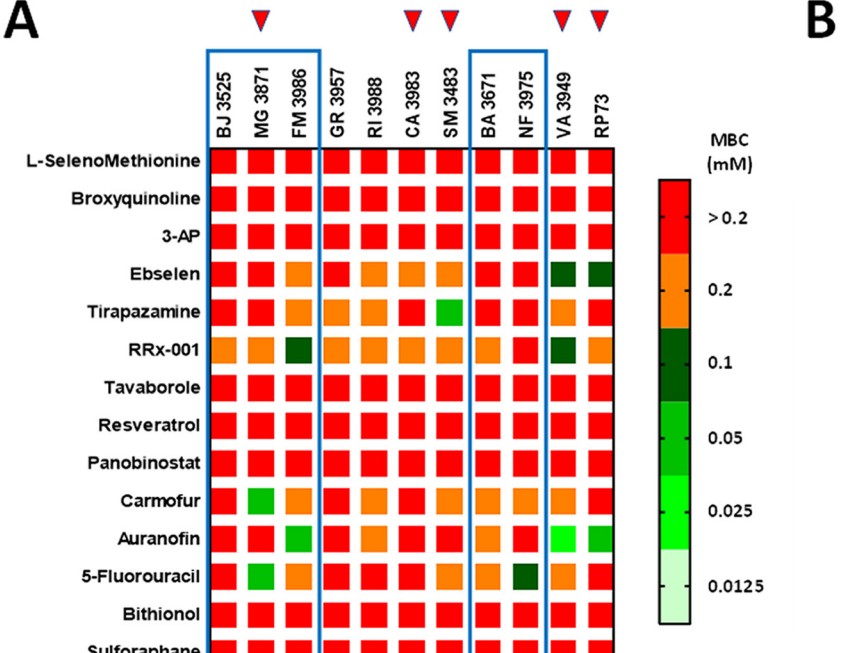

**FIG 4** *In vitro* activity of selected antibacterial hits against *P. aeruginosa* strains from CF patients. (A) Heat map of MBC data (millimolar concentration) for the tested hit compounds. The values, expressed as millimolar concentrations, are color-coded, and color bars mark the matrix positions of compounds in a particular *P. aeruginosa* strain. Boxed are strains with strong pyocyanin, pyoverdine and protease production, while red triangles indicate MDR strains. (B) Lowest concentrations bactericidal for 50% and 90% of the tested strains ($MBC_{50}$ and $MBC_{90}$, respectively), expressed as millimolar concentrations (micrograms per milliliter).

Auranofin was the most active hit ($MIC_{50}$, 0.025 mM; $MIC_{90}$, 0.1 mM), followed by 5-fluorouracil ($MIC_{50}$, 0.025 mM; $MIC_{90}$, 0.2 mM), tirapazamine ($MIC_{50}$, 0.05 mM; $MIC_{90}$, 0.2 mM), carmofur and RRx-001 ($MIC_{50}$, 0.1 mM; $MIC_{90}$, 0.2 mM) (Fig. 3). Auranofin, carmofur, and RRx-001 showed MICs of ≤0.1 mM against all MDR strains (MG3871, CA3983, SM3483, VA3949, RP73), suggesting that the development of cross-resistance with the antibiotics is unlikely to occur. On the other hand, 5-fluorouracil showed high activity against strains classified as strong producers of pyocyanin, pyoverdine, and protease (BA3671, BJ3525, FM3986, NF3975, MG3871) (Fig. 3). Resveratrol, panobinostat, bithionol, and sulforaphane were the least active, showing $MIC_{50}$s and $MIC_{90}$s of >0.2 mM.

$MBC_{50}$ values did not allow comparative analysis since they were less variable, ranging from 0.2 to >0.2 mM, while the $MBC_{90}$ was always higher than 0.2 mM (Fig. 4). An antimicrobial compound is typically considered bactericidal if the MBC does not exceed 4× MIC (23). Considering the measurable MBC/MIC ratio values, it was possible to infer that RRx-001 exerts bactericidal activity against most strains tested (9 out of 11), followed by ebselen (6 out of 11), tirapazamine (5 out of 11), carmofur (4 out of 11), auranofin (3 out of 11), and 5-fluorouracil (2 out of 11). No bactericidal effect could be observed for other hits due to a ratio of >4 or out-of-range MBC values.

No correlation could be proved between the sensitivity of the tested strains to the hit compounds and the presence of the antibiotic resistance genes or infection statuses (Table S2). Interestingly, RP73 was the most susceptible strain among those tested, thus highlighting the importance of confirming the activity of a potential hit against a comprehensive set of strains. Further work is warranted to explore the mechanisms underlying the strain-specific activity seen among the hits, including the target of the compounds.

**Cytotoxic potential.** Repurposing existing drugs offers the advantage of known pharmacokinetic profiles and safety. However, for novel applications, the cytotoxic effects of these compounds are still to be investigated. Considering a potential therapeutic application of the 14 hits selected from the primary antibacterial screening, we

examined their possible cytotoxic effect on IB3-1 bronchial epithelial cells isolated from a pediatric CF patient using an MTS [3-(4,5-dimethylthiazol-2-yl)-5-(3-carboxyme-thoxyphenyl)-2-(4-sulfophenyl)-2H-tetrazolium]-based colorimetric assay.

Tavaborole showed the safest profile since exposure for 24 h to concentrations until 0.8 mM was not toxic to IB3-1 cells, followed by L-selenomethionine, 5-fluorouracil, panobi-nostat, and resveratrol, which were not toxic until 0.4 mM (see Fig. S1 in the supplemental material). Conversely, bithionol and auranofin were the most toxic to IB3-1 cells, causing their death already at concetnrations of 0.05 and 0.0125 mM, respectively (Fig. S1).

Fifty percent lethal concentration ($LC_{50}$) values, calculated after absorbance kinetic data normalization and nonlinear regression, showed a comparable pattern, with tava-borole as the less toxic hit (1.158), followed by L-selenomethionine (1.066) and 5-fluo-rouracil (1.061) (Fig. 5). The other hits showed $LC_{50}$ values of $<1$, although to different extents. Particularly, bithionol, RRx-01, and auranofin resulted in the most toxic hits, with values of 0.098, 0.070, and 0.009, respectively (Fig. 5).

Next, as a preliminary view of a compound's therapeutic index, we calculated the ratios between the highest hit concentration not toxic to IB3-1 cells and the corresponding $MIC_{50}$ and $MIC_{90}$ values (Table 2). In terms of the maximum noncytotoxic concentration (MNCC)/$MIC_{50}$ ratios, 5-fluorouracil showed the highest ratio, followed by tavaborole (ratios of 16 and 8, respectively), while L-selenomethionine, ebselen, tirapazamine, and carmofur had a value of 2. The worst ratio values of $\leq 1$ were associated with RRx-001, resveratrol, and panobino-stat. A significant decrease in values was observed when the MNCC/$MIC_{90}$ ratio is consid-ered. 5-Fluorouracil showed the highest value, followed by L-selenomethionine, and car-mofur (ratios of 2, $\geq 1$, and 1, respectively). The remaining hits showed ratios of $<1$, with the worst values of $\leq 0.0625$ associated with auranofin and bithionol. Tobramycin showed the highest MNCC/$MIC_{50}$ and MNCC/$MIC_{90}$ values (64 and 32, respectively).

Taken together, our findings showed that the therapeutic index is marginal. However, it is worth noting that 5-fluorouracil, the safest among the hits, showed a high MNCC/$MIC_{50}$ ratio, only $2\text{-}log_2$ lower than that calculated for tobramycin, which would make it possible to create an aerosolized formulation to achieve higher concentrations and increase lung penetration, which is especially important for CF pulmonary infections.

Based on the findings from cytotoxicity and MIC assays, only the hits showing the poten-tial for clinical use—i.e., those with a MIC of $\leq 0.1$ mM against at least 50% of *P. aeruginosa* strains tested, and with a $MIC_{50}$ not cytotoxic to IB3-1 cells—were selected for further charac-terization: ebselen, tirapazamine, carmofur, tavaborole, and 5-fluorouracil.

**Time-kill assay.** To investigate the bactericidal activity of the selected five hits, time-kill assays were compared to the commonly used antibiotic tobramycin (Fig. 6). The results confirmed the predicted antimicrobial effect based on the MBC/MIC values.

Specifically, a rapid and dose-dependent bactericidal effect was observed when *P. aeruginosa* RP73 cells were exposed for 2 h and 4 h, respectively, at $4\times$ and $2\times$ MIC of ebselen. However, the cell count decreased below the detection limit without regrowth over 24 h only at $4\times$ MIC, while a reduction by 4.3 $log_{10}$ CFU/mL with a later trend to regrowth was seen after exposure at $2\times$ MIC. Other hits showed bacteriostatic activity regardless of the concentration tested, with carmofur as the most active causing, after 24 h of exposure at $4\times$ MIC, a reduction in cell counts by 2.6 $log_{10}$ CFU/mL. Exposure to tavaborole at all tested con-centrations allowed *P. aeruginosa* growth similar to that of unexposed samples. Tobramycin showed a precise dose- and time-dependent effect resulting in a bactericidal effect irrespective of the concentration tested. At $4\times$ MIC, the cell count settled below the detection limit more slowly than ebselen did (6 h versus 2 h, respectively).

Overall, our findings showed that only ebselen has the potential to cause rapid and dose-dependent bactericidal activity against *P. aeruginosa*. It is plausible to infer that ebse-len retains this effect at least against 50% of *P. aeruginosa* CF strains tested, as suggested by the MBC/MIC ratio of 2.

**Drug combination assay.** Combination therapy is an important alternative when monotherapy is not effective. Indeed, synergism between two agents may reduce treatment toxicity by administering lower doses and reducing antimicrobial resistance. Few studies have tested anticancer-antibiotic combinations for synergistic activity (24).

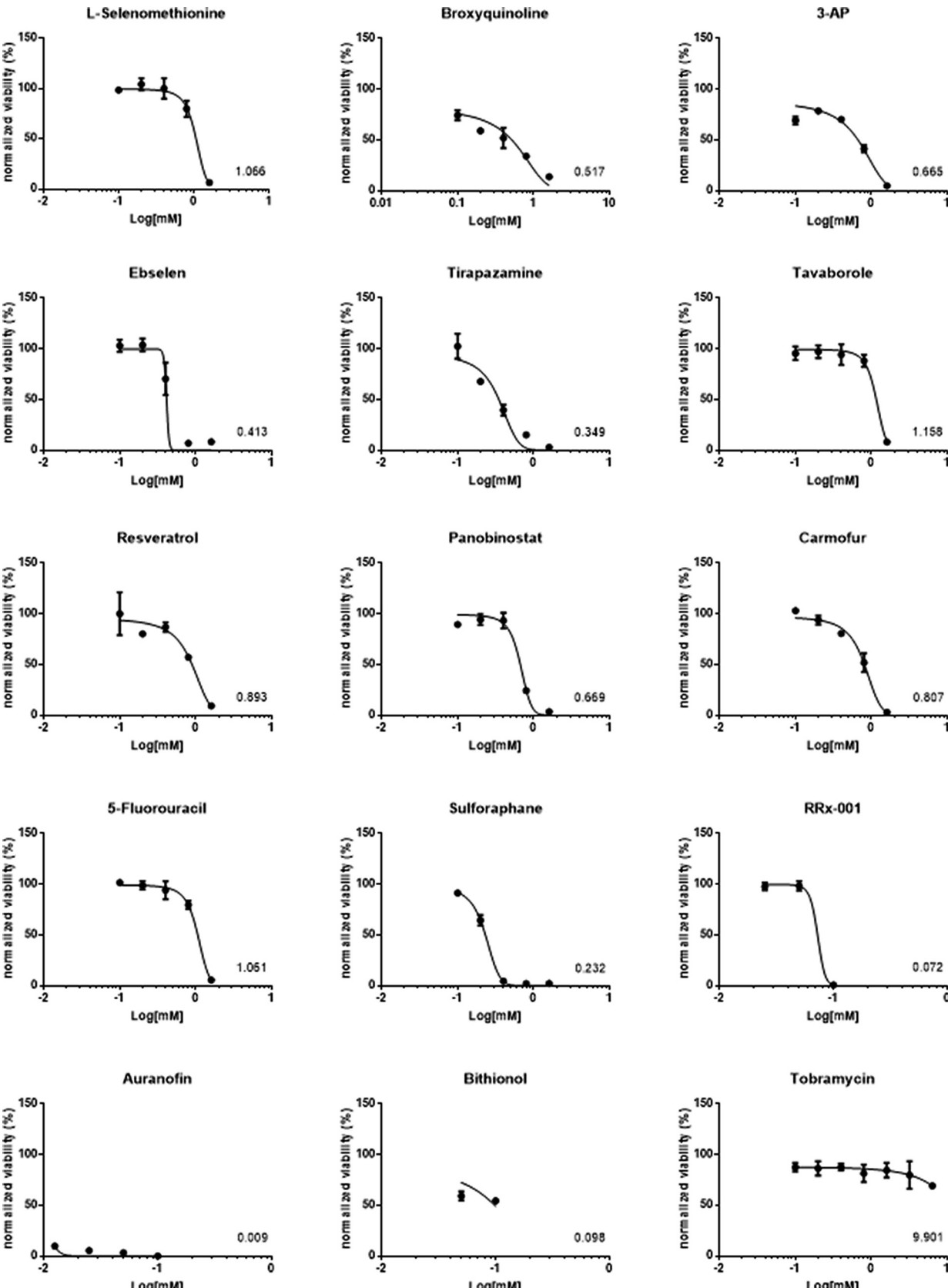

**FIG 5** Dose-response curves of the cytotoxic activity of each hit. IB3-1 bronchial epithelial cell monolayers were exposed for 24 h to several hit concentrations, and then cell viability was assessed using an MTS-based colorimetric assay. The kinetic absorbance data were normalized using the positive-control absorbance value as 100% viability. The log value of the concentration required to kill cells by 50% ($LC_{50}$) compared to the control is indicated. Data points represent mean values, and error bars represent the standard deviation (SD) ($n = 6$).

**TABLE 2** Potential therapeutic application of the 14 hits selected from the findings of the primary HTS[a]

| Antibacterial hit | MNCC (mM) | $MIC_{50}$ (mM) | $MNCC/MIC_{50}$ ratio | $MIC_{90}$ (mM) | $MNCC/MIC_{90}$ ratio |
|---|---|---|---|---|---|
| L-Selenomethionine | 0.4 | 0.2 | 2 | >0.2 | ≥1 |
| Broxyquinoline | <0.1 | 0.2 | ≤0.25 | >0.2 | ≤0.125 |
| 3-AP | <0.1 | 0.2 | ≤0.25 | >0.2 | ≤0.125 |
| Ebselen | 0.2 | 0.1 | 2 | >0.2 | ≤0.5 |
| Tirapazamine | 0.1 | 0.05 | 2 | 0.2 | 0.5 |
| RRx-001 | 0.05 | 0.1 | 0.5 | 0.2 | 0.25 |
| Tavaborole | 0.8 | 0.1 | 8 | >0.2 | ≤0.5 |
| Resveratrol | 0.4 | >0.2 | ≤1 | >0.2 | ≤1 |
| Panobinostat | 0.4 | >0.2 | ≤1 | >0.2 | ≤1 |
| Carmofur | 0.2 | 0.1 | 2 | 0.2 | 1 |
| Auranofin | <0.0125 | 0.025 | ≤0.25 | 0.1 | ≤0.0625 |
| 5-Fluorouracil | 0.4 | 0.025 | 16 | 0.2 | 2 |
| Bithionol | <0.05 | >0.2 | ≤0.06 | >0.2 | ≤0.0625 |
| Sulforaphane | <0.1 | >0.2 | ≤0.125 | >0.2 | ≤0.125 |
| Tobramycin | 3.2 | 0.05 | 64 | 0.1 | 32 |

[a]For each hit, the ratio between the maximum hit concentration not toxic to IB3-1 cells (maximum noncytotoxic concentration [MNCC]) and the corresponding $MIC_{50}$ value or $MIC_{90}$ value was calculated.

5-Fluorouracil was reported to interact synergistically with $\beta$-lactam antibiotics when tested against Gram-negative bacilli, such as *P. aeruginosa*, *Klebsiella pneumoniae*, and *Proteus mirabilis* (25, 26).

To evaluate whether the selected hits could synergistically interact with tobramycin, we performed a microdilution checkerboard combination assay against the *P. aeruginosa* RP73 tobramycin-resistant strain, and fractional inhibitory concentration index (FICI) values were calculated. No synergistic interaction was found, as suggested by the FICI values ranging from 1.06 to 2.00 (data not shown), confirming that the hits have antibacterial activity on their own. Further studies on other drugs approved for CF, such as levofloxacin and colistin, are warranted.

**Antibiofilm activity against *P. aeruginosa* biofilm.** The ability to form intrinsic resistant biofilms by *P. aeruginosa* is a relevant mechanism underlying persistent infection in the lung of CF patients (27). Therefore, the selected hits were evaluated using a crystal violet assay for their potential to affect biofilm formation and cause the dispersion of preformed biofilm by *P. aeruginosa* RP73 (Fig. 7 and 8).

All the hits tested caused a statistically significant reduction in biofilm formation compared to the control after 24 h, although to different extents (Fig. 7). Carmofur and

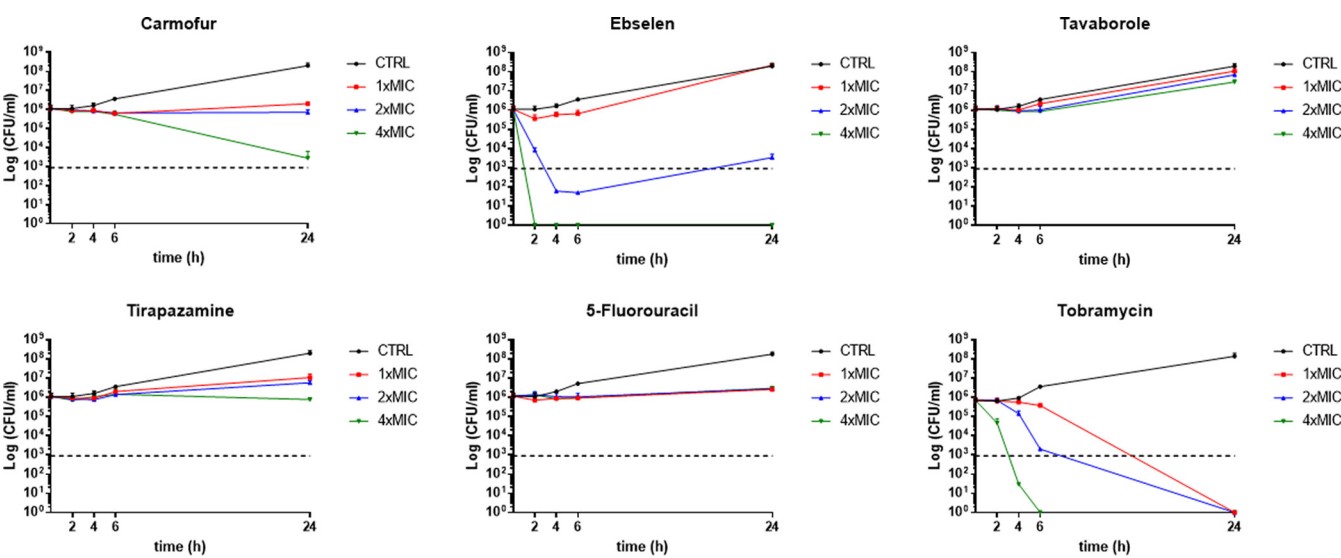

**FIG 6** Time-kill kinetics against *P. aeruginosa* RP73 strain. Each drug was tested at the MIC value (0.05 mM for all hits; 128 $\mu$g/mL for tobramycin) and 2× and 4× MIC. The dotted line indicates the bactericidal activity, defined as a ≥3-log (CFU/mL) reduction of the initial inoculum size. The limit of detection was 10 CFU/mL. CTRL, control. Results are mean values + SD (*n* = 6); SD bars were too small to be visualized on the graph in several cases.

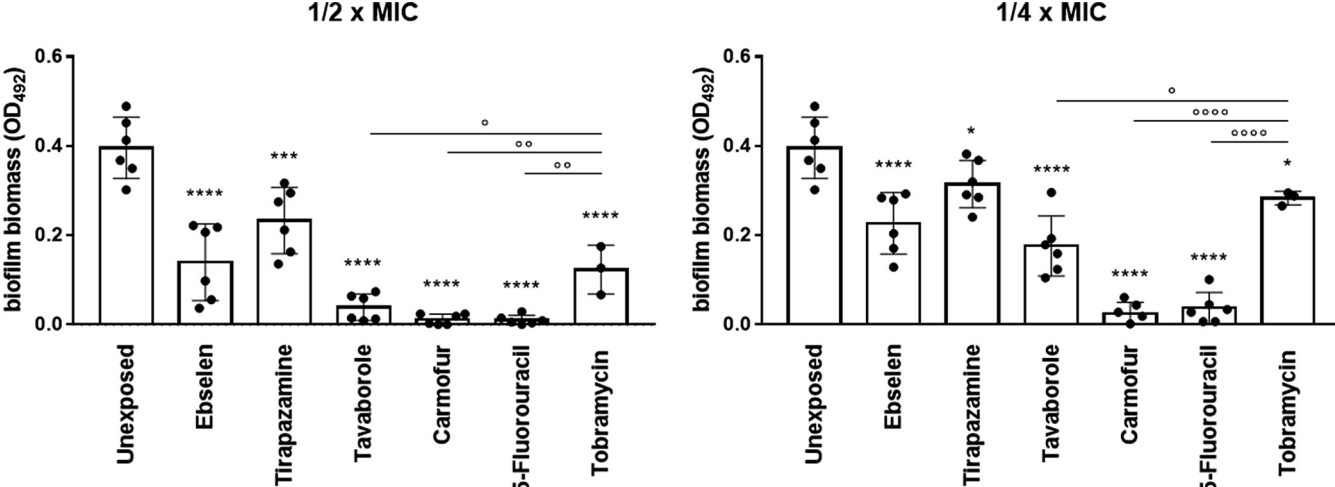

**FIG 7** *In vitro* activity of the selected hits against biofilm formation by the *P. aeruginosa* RP73 strain. Biofilm was allowed to grow for 24 h in the presence of each hit or tobramycin at subinhibitory concentrations (1/2× and 1/4× MIC), and biofilm biomass was measured by crystal violet stain assay. Unexposed control samples were not exposed. Results are shown as individual data points and mean values ± SD ($n = 6$). Statistical significance: *, $P < 0.05$, ***, $P < 0.001$, and ****, $P < 0.0001$, versus unexposed sample, by one-way ANOVA plus Dunnett's multiple-comparison posttest; °, $P < 0.05$, °°, $P < 0.01$, and °°°°, $P < 0.0001$, by unpaired *t* test.

5-fluorouracil were the most active drugs, preventing biofilm formation over the tested concentration range. Tirapazamine was the least active, reducing biofilm biomass by 25.3% and 41.1%, respectively, at 1/4× and 1/2× MIC. The antibiofilm activity was dose dependent in the case of ebselen, tirapazamine, and tavaborole. To our knowledge, this is the first evidence of antibiofilm activity reported for tavaborole and tirapazamine. About ebselen, although it has been reported to significantly affect biofilm formation by *Helicobacter pylori* (28), vancomycin-resistant enterococci (29), and staphylococci (30), no studies have been conducted on *P. aeruginosa*. The antibiofilm effect we saw in the present study could be due to the inhibition of alginate synthesis (31). In agreement with our findings, Guendoze et al. found that 5-fluorouracil, acting as an inhibitor of quorum sensing (QS), affects biofilm formation in *P. aeruginosa*, although at concentrations significantly higher than those seen in the present study ($\geq 60$ $\mu$M versus 12.5 $\mu$M, respectively) (32). Other studies also reported that 5-fluorouracil efficiently reduces

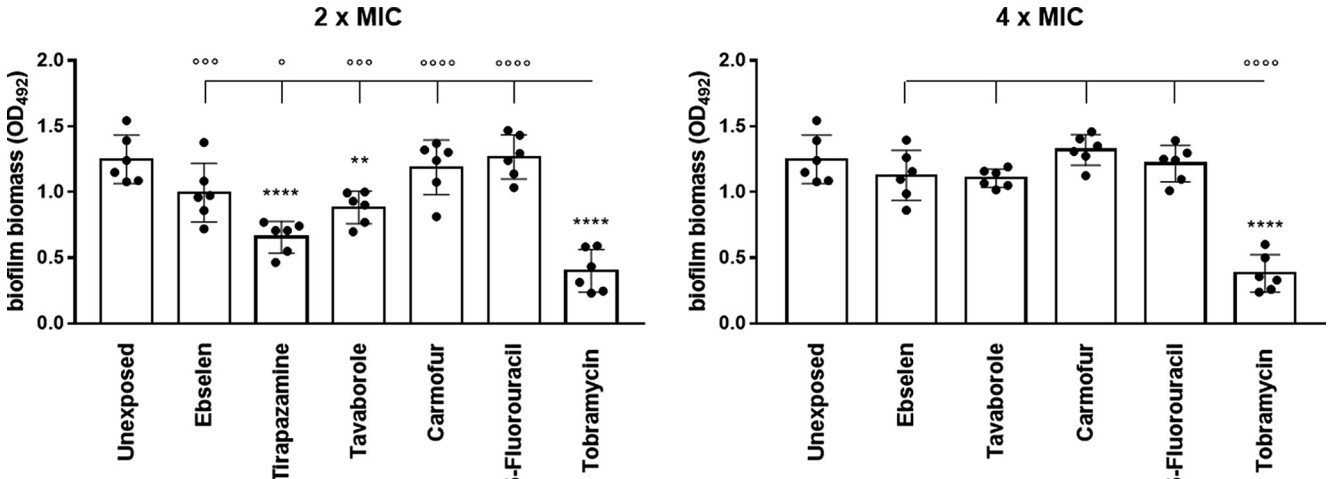

**FIG 8** *In vitro* activity of the selected hits against biofilm preformed by the *P. aeruginosa* RP73 strain. Biofilm grown for 24 h was exposed to each hit or tobramycin at 2× and 4× MIC, and after 24 h of exposure, biofilm biomass was measured by crystal violet stain assay. Tirapazamine was not tested at 4× MIC due to its toxicity toward IB3-1 cells. Unexposed control samples were not exposed. Results are shown as individual data points and mean values ± SD ($n = 6$). Statistical significance: **, $P < 0.01$, and ****, $P < 0.0001$, versus unexposed sample, by one-way ANOVA plus Dunnett's multiple-comparison posttest; °, $P < 0.05$, °°°, $P < 0.001$, and °°°°, $P < 0.0001$, by unpaired *t* test.

adherence and blocks biofilm formation of methicillin-resistant *S. aureus* (MRSA) and *Escherichia coli* by inhibiting quorum-sensor autoinducer-2 production (33) and through global regulator AriR (34), respectively. Another mechanism could be the downregulation of expression of *algD* coding for GDP-mannose 6-dehydrogenase involved in the alginate polymerization needed for adhesion and following biofilm formation (35). From a translational point of view, clinical trials in humans testing the effect of 5-fluorouracil as an antibiofilm external coating of central venous catheters in critically ill patients have also shown its efficacy in abolishing colonization by Gram-negative bacteria and preventing bloodstream infections (36). Tobramycin significantly affected biofilm formation in a dose-dependent manner (40.4 and 68.2% reductions versus the control, respectively, at $1/4\times$ and $1/2\times$ MIC), although with lower efficacy than 5-fluorouracil, carmofur, and tavaborole (*P* values of at least $<0.05$).

Next, for the first time in the literature, we assessed the effect of the five hits against mature biofilm formed by *P. aeruginosa* RP73 after 24 h of exposure at multiples of the corresponding MIC value (Fig. 8). Tobramycin was the most active hit, causing a significant, dose-independent, reduction of biofilm biomass (76.1 and 72.1% versus the control, respectively, at $2\times$ and $4\times$ MIC). Among hits, tirapazamine and tavaborole were the only active drugs, showing a comparable ability to disperse preformed biofilm. Specifically, when tested at $2\times$ MIC, biofilm reductions of 48.2% and 43.1%, respectively, were observed compared to the unexposed control.

Taken together, our findings showed that all selected hits have the potential to be used in the early, aggressive therapy to prevent, or at least delay, biofilm formation even more efficiently than tobramycin, as in the cases of 5-fluorouracil, carmofur, and tavaborole. The effect on biofilm formation is not dependent on antibacterial activity since they were tested at subinhibitory concentrations. On the other hand, tavaborole and tirapazamine could be relevant to treating established, chronic infection by *P. aeruginosa* due to their specific effectiveness in disrupting the preformed biofilms.

**In vitro activity against other major respiratory pathogens in CF.** In CF patients, the modification of the pulmonary environment and the alteration of mucociliary clearance favor the colonization of specific bacterial and fungal species that can interact synergistically or antagonistically (37, 38). Besides changing the composition of the pulmonary microbiota and the adaptation and virulence of specific pathogens, these interactions can have clinical consequences affecting antibiotic treatments for respiratory infections.

In this framework, we sought to investigate the potential broader-range applicability of the anti-*P. aeruginosa* hits by assessing their activity, in terms of MIC and MBC, against a panel of strains representative of other pulmonary pathogens relevant to CF patients, namely, *Staphylococcus aureus*, *Stenotrophomonas maltophilia*, *Burkholderia cepacia*, and *Acinetobacter baumannii* (Fig. 9).

*S. aureus* was the most susceptible species to the hits tested, whose MIC was $\leq 0.1$ mM for 15 out of 18 strain-hit combinations (83.3%), followed by *B. cepacia* (12/18 [66.7%]), *S. maltophilia* (5/18 [27.8%]), and *A. baumannii* (2/18 [11.1%]). MBC values were 0.2 mM or higher, except for tirapazamine versus the Sa16 strain (MBC of 0.1 mM). The apparent reasons for these findings could be the structural and molecular differences among these species. Specifically, the presence of an outer membrane may serve as a barrier for the diffusion of a compound to the cell wall and cytoplasm, thus explaining the higher resistance observed among Gram-negative species.

When the hits were comparatively evaluated, tavaborole was the most active (MIC of $\leq 0.1$ mM against 6 out of 8 strains tested), especially against *B. cepacia* and *A. baumannii* strains, followed by carmofur (5 out of 8 strains) and ebselen and tirapazamine (4 out of 8 strains), which were particularly active against *S. aureus* and *B. cepacia* strains. 5-Fluorouracil was the least active, showing MICs of $\leq 0.1$ mM only against both *S. aureus* strains. Tobramycin was significantly less potent than the tested hits, showing MIC and MBC values of $>0.2$ mM.

Overall, our findings showed that tavaborole and carmofur have broad-spectrum bactericidal activity against CF pathogens, indicating their potential as a therapeutic possibility for CF patients. Further studies are warranted to explore the mechanisms underlying the species-specific activity of the hits.

| | S. aureus | | | | A. baumannii | | | | S. maltophilia | | | | B. cepacia | | | |
|---|---|---|---|---|---|---|---|---|---|---|---|---|---|---|---|---|
| | Sa2 | | Sa16 | | Ab63 | | Ab219 | | SanG | | Sm142 | | Bc11 | | Bc23 | |
| | MIC | MBC | MIC | MBC | MIC | MBC | MIC | MBC | MIC | MBC | MIC | MBC | MIC | MBC | MIC | MBC |
| L-SelenoMethionine | 0.05 | >0.2 | >0.2 | >0.2 | >0.2 | >0.2 | >0.2 | >0.2 | 0.0125 | >0.2 | 0.025 | >0.2 | 0.025 | >0.2 | 0.1 | >0.2 |
| Ebselen | 0.1 | 0.2 | 0.05 | >0.2 | 0.2 | >0.2 | 0.2 | >0.2 | 0.2 | 0.2 | >0.2 | >0.2 | 0.025 | 0.2 | 0.025 | 0.2 |
| Tirapazamine | 0.05 | 0.2 | 0.05 | 0.1 | 0.2 | 0.2 | 0.2 | >0.2 | >0.2 | >0.2 | >0.2 | >0.2 | 0.1 | >0.2 | 0.05 | 0.2 |
| Tavaborole | 0.05 | >0.2 | 0.2 | >0.2 | 0.025 | >0.2 | 0.05 | >0.2 | 0.1 | >0.2 | 0.2 | >0.2 | 0.025 | >0.2 | 0.025 | >0.2 |
| Resveratrol | 0.05 | >0.2 | 0.2 | >0.2 | >0.2 | >0.2 | >0.2 | >0.2 | 0.2 | >0.2 | 0.2 | >0.2 | 0.2 | >0.2 | 0.1 | >0.2 |
| Panobinostat | 0.025 | >0.2 | 0.025 | >0.2 | >0.2 | >0.2 | 0.2 | >0.2 | >0.2 | >0.2 | >0.2 | >0.2 | 0.2 | >0.2 | 0.2 | >0.2 |
| Carmofur | 0.025 | >0.2 | 0.1 | >0.2 | >0.2 | >0.2 | 0.2 | >0.2 | 0.025 | >0.2 | 0.2 | >0.2 | 0.025 | 0.2 | 0.025 | >0.2 |
| 5-Fluorouracil | 0.025 | >0.2 | 0.05 | >0.2 | >0.2 | >0.2 | >0.2 | >0.2 | 0.2 | >0.2 | 0.2 | >0.2 | 0.2 | >0.2 | 0.2 | >0.2 |
| RRx-001 | 0.1 | >0.2 | 0.1 | 0.2 | 0.2 | 0.2 | 0.2 | 0.2 | 0.2 | >0.2 | 0.1 | >0.2 | 0.05 | >0.2 | 0.2 | >0.2 |
| Tobramycin | >0.2 | >0.2 | >0.2 | >0.2 | >0.2 | >0.2 | >0.2 | >0.2 | >0.2 | >0.2 | >0.2 | >0.2 | >0.2 | >0.2 | >0.2 | >0.2 |

**FIG 9** *In vitro* activity of the selected antibacterial hits and tobramycin against some CF strains representative of species other than *P. aeruginosa*, namely, *S. aureus*, *S. maltophilia*, *B. cepacia*, and *A. baumannii*. The MIC and MBC were measured using the broth microdilution method and expressed as millimolar concentration. Values highlighted in green are MICs of ≤0.1 mM.

**Antibacterial mechanism.** To gain insights into the mechanism(s) underlying the activity of the hits against *P. aeruginosa* RP73, we first conducted a focused-ion-beam scanning electron microscopy (FIB/SEM) analysis to visually observe the effects on the cell morphology and ultrastructure after overnight exposure to the selected hits at 2× MIC (Fig. 10A). Contrary to the unexposed bacteria, showing unaltered morphology with regular contours of outer and inner membranes, cells exposed to hits exhibited features suggestive of membrane damage, although to different extents. Ebselen caused surface blistering due to the development of regular and rounded vesicular structures. Most cells appeared strongly electron dense with the inner membrane scarcely distinguishable, while the outer membrane remained discernible and unaffected. A minority of bacteria showed the total loss of intracellular content, indented outer membrane, irregular/shrunken shape, and uneven or absent cytoplasmatic electron density. Ebselen-specific effects were observed in the intercellular areas with strongly electron-dense filamentous material and bigger aggregates of less electron-dense material. Cells exposed to carmofur displayed a distinctive morphological alteration involving swelling at the polar regions secondary to an increased periplasmic space, which, however, did not cause any

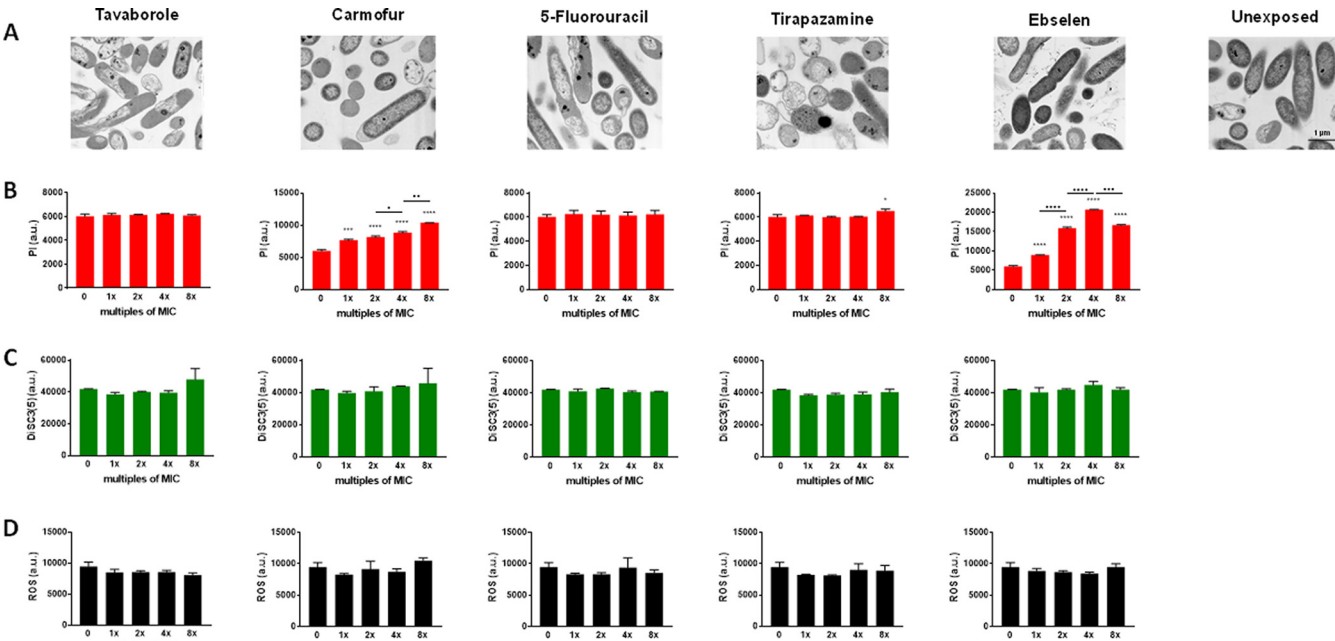

**FIG 10** Mechanism of action underlying the antibacterial effect of hits against *P. aeruginosa* RP73. (A) FIB/SEM analysis representative images of *P. aeruginosa* RP73 exposed to 2× MIC of each hit. All images were taken at the same magnification; the scale bar represents 1 $\mu$m and is indicated in the "unexposed" image. (B) Propidium iodide (PI) uptake after 2 h of exposure to each hit; (C) detection of membrane depolarization with DiSC3(5) probe measured immediately after exposure; (D) total reactive oxygen species (ROS) accumulation after 6 h of treatment with each hit. Results are shown as mean values + SD ($n = 6$) of fluorescence intensity in arbitrary units (a.u.). Statistical significance was evaluated by ordinary one-way ANOVA followed by Dunnett's (\*, $P < 0.05$, \*\*\*, $P < 0.001$, and \*\*\*\*, $P < 0.0001$, versus unexposed cells) or Tukey's (°, $P < 0.05$, °°, $P < 0.01$, °°°, $P < 0.001$, and °°°°, $P < 0.0001$) multiple-comparison tests.

rupture of the inner and outer membranes. Cells treated with tirapazamine showed outer membrane corrugation and many cells with residual or absent cytoplasm, and swelling phenomena could also be rarely reported. Exposure to 5-fluorouracil caused perforations and clear membrane rupture more frequently than other hits. Wrinkling, shrinkage, and content loss were also observed. A blistering phenomenon was also found with the formation of vesicles, although it occurs less frequently than with ebselen. Bacteria treated with tavaborole often showed leakage and partial loss of cytoplasmic contents; shrinkage and other outer membrane alterations were also commonly observed, while mild swelling and outer membrane vesicles were rarely detected.

To assess the causes of membrane damage observed at microscopic analysis, we evaluated possible membrane permeability changes. Propidium iodide (PI) is a fluorescent probe able to penetrate the damaged cell membrane and intercalate to DNA and was, therefore, used to investigate the permeability of the cell membrane. The exposure to carmofur, ebselen, and tirapazamine increased fluorescence intensity, although to different extents. Specifically, ebselen and carmofur were effective at each concentration tested, revealing a dose-dependent effect (ebselen, $1\times$ versus $2\times$ MIC, $P < 0.0001$; $2\times$ versus $4\times$ MIC, $P < 0.0001$; $4\times$ versus $8\times$ MIC, $P < 0.001$; carmofur, $1\times$ versus $2\times$ MIC, $P > 0.05$; $2\times$ versus $4\times$ MIC, $P < 0.05$; $4\times$ versus $8\times$ MIC, $P < 0.01$) (Fig. 10B). The significant decrease in fluorescence observed when ebselen was tested at $8\times$ MIC was probably due to the relevant cell lysis, as inferable from time-kill kinetics findings (Fig. 6). Tirapazamine was active only at $8\times$ MIC, while tavaborole and 5-fluorouracil never increased the fluorescence over the concentration range tested.

Changes in the permeability of the cell membrane can reflect changes in its conductivity (39). Therefore, we used the fluorescent dye DiSC3(5), particularly sensitive to membrane potential changes, to observe the depolarization effects. We found that fluorescent intensity was not significantly different with versus without exposure to hits, regardless of the concentration tested (Fig. 10C), thus suggesting that the membrane damage caused by the hits is unrelated to changes in membrane potential.

PI uptake and depolarization assays indicated that the membrane damage caused by ebselen and carmofur, and to a lesser extent by tirapazamine, might be related to increased membrane permeability. Conversely, in the cases of 5-fluorouracil and tavaborole, the disrupted appearance on FIB/SEM images and leakage of intracellular material might be due to other impairments than membrane destruction. In this framework, the inconsistency between FIB/SEM analysis and those revealed by the PI uptake assay could be explained by the prolonged exposure time used for microscopic analysis and by other factors, among those the presence of damaged DNA. Indeed, as PI intercalates with intact DNA, the lack of staining bacteria could hint at the breakage of DNA after tratment with the hit, which may be caused by the formation of ROS (29, 40–42). Therefore, we measured the ROS formation through H2DCFDA (2′,7′-dichlorodihydrofluorescein diacetate), a nonfluorescent dye converted into the fluorescent dye DCF in the presence of intracellular ROS. We did not find any increase in fluorescent intensity in bacterial cells exposed to hits, regardless of the concentration tested (Fig. 10D), thus indicating that induced ROS formation is not a mechanism underlying the antibacterial effects of 5-fluorouracil and tavaborole or the other hits.

Overall, our findings indicated that ebselen, carmofur, and, at high concentrations, tirapazamine exert their antibacterial activity mainly by targeting the cell membrane, increasing its permeability with subsequent cell lysis. Further work is warranted to explore the attractive properties associated with membrane-targeting antimicrobials (e.g., low prospect for resistance selection, synergism with other antibiotics, antipersister potential) (43), and to identify the precise molecular mechanisms responsible for the activity of 5-fluorouracil and tavaborole.

**Selected repositionable candidate hits: comparison with antibacterial data from the literature.** In the present study, the five most promising molecules were revealed by the HTS performed, for the first time, under "CF-like" experimental conditions.

**(i) Ebselen.** Ebselen [*N*-phenyl-1,2-benzisoselenazol-3(2H)-one], one of the organoselenium compounds that mimic glutathione peroxidase, shows a wide range of biological activities (44). Although not yet FDA approved, ebselen is safe in humans and is being

evaluated in phase III trials for bipolar disorder, the treatment and prevention of hearing loss, Meniere's disease, and severe acute respiratory syndrome coronavirus 2 (SARS-CoV-2) (44–46). *In vitro* studies revealed antibacterial activity—targeting the thioredoxin reductase, thus leading to an intracellular elevation of ROS—for a wide range of infectious pathogens, including MDR Gram-positive bacteria, *A. baumannii*, and *Mycobacterium tuberculosis* complex (29, 40–42). Our findings showed that ebselen exerts activity toward *P. aeruginosa* by damaging the membrane cell due to augmented membrane permeability rather than increased production of intracellular ROS. Very few studies on *P. aeruginosa* reported an activity significantly lower than we observed, with MIC values at least 4-fold higher than the $MIC_{50}$ measured in the present study (47, 48). In addition, our findings showed that ebselen, contrary to other hits, can potentially cause rapid bactericidal activity against *P. aeruginosa*. Ebselen also has various anti-inflammatory properties that could be relevant in CF patients where chronic inflammation causes lung tissue damage, eventually leading to respiratory failure. Indeed, in a rat model of lipopolysaccharide (LPS)-induced pulmonary inflammation, ebselen attenuates neutrophil recruitment and activation by decreasing lung ICAM-1 expression secondary to the inhibition of tumor necrosis factor alpha (TNF-$\alpha$) and interleukin-1$\beta$ (IL-1$\beta$) (49). In addition, mimicking the effect of glutathione peroxidase, it neutralizes ROS by the glutathione (GSH)-peroxidase-coupled reaction, thereby blocking the pathway to cytokine enhancement (50).

**(ii) Tirapazamine.** Tirapazamine (3-amino-1,2,4-benzotriazine 1,4 dioxide) is an anticancer agent showing selective cytotoxicity for hypoxic cells in solid tumors. Upon its activation, this aziridinylquinone forms a free radical at position 1, where the oxide group is attached, inducing single- and double-strand breaks in DNA, base damage, and cell death. Previous studies reported its activity against *E. coli*, *Salmonella enterica*, *S. aureus*, *Clostridium difficile*, and *M. tuberculosis* (51–53). In the present study, for the first time, we observed the effectiveness of tirapazamine against *P. aeruginosa*, not only against planktonic cells but also in dispersing preformed biofilm.

In addition, it is worth noting that tirapazamine is a prodrug activated under hypoxic conditions (21), such as those found not only inside solid tumors—where $O_2$ levels are generally low due to poor vascularization—but also in CF lung, where chronic infection leads to a considerably reduced $O_2$ tension (10, 12).

**(iii) Carmofur.** Carmofur (1-hexylcarbamoyl-5-fluorouracil) is an antimetabolite (pyrimidine analog) antineoplastic derivative of 5-fluorouracil used in the treatment of breast and colorectal cancer. As far as we know, only one study has reported the antibacterial activity of carmofur. Peyclit et al. (54), screening an FDA-approved drug library against highly resistant bacteria in CLSI-recommended cation-adjusted Mueller-Hinton broth, reported carmofur was active against *S. aureus* and *Enterococcus faecium*, but not against the Gram-negative bacteria *K. pneumoniae*, *A. baumannii*, and *P. aeruginosa*. On the contrary, herein for the first time, under CF-like conditions we found carmofur active against planktonic cells and in preventing biofilm formation by *P. aeruginosa* cells, thus highlighting the importance of considering physicochemical conditions relevant to the site of infection.

Carmofur also exhibits properties potentially relevant to CF patients, and drug repurposing may provide us with effective therapeutic strategies for CF lung infections. Indeed, in a recent study carried out in C57BL/6J mice with LPS-induced acute lung injury, carmofur significantly reduced inflammation and neutrophil infiltration through the NF-$\kappa$B signaling pathway; in addition, after oral administration, it was stable in the lung and suitable for systemic administration (55).

**(iv) Tavaborole.** Tavaborole (5-fluorobenzoxaborole) is an antifungal agent approved by the FDA as a 5% (wt/vol) topical solution formulation for the potential treatment of toenail onychomycosis caused by *Trichophyton* species. It inhibits cytoplasmic leucyl-transfer RNA synthetase, thus interfering with fungal protein synthesis (56). Because aminoacyl-tRNA synthetases widely exist in all organisms, they might be an ideal target for discovering new synthetic antimicrobial drugs. Several studies have shown that tavaborole's chemical derivatives are active against bacterial pathogens, including *S. aureus*, *M. tuberculosis*, *Streptococcus pneumoniae*, and MDR strains of *P. aeruginosa* and *E. coli* (57). However,

our findings for the first time revealed anti-*P. aeruginosa* activity of tavaborole against both planktonic cells and preformed biofilms.

**(v) 5-Fluorouracil.** 5-Fluorouracil is an analog of uracil widely used to treat several cancers: systematically for colorectal, oesophageal, stomach, anal, breast, pancreatic, head, and neck cancers or topically for skin cancers and actinic keratoses. It affects pyrimidine synthesis by inhibiting thymidylate synthetase, leading to cell death due to the inhibition of DNA and RNA synthesis. Fluorouracil can also be incorporated into RNA instead of uridine triphosphate, producing a fraudulent RNA and interfering with RNA processing and protein synthesis. Based on its broad effects against cancer cells, it is not surprising that 5-fluorouracil also has a broad-spectrum antibacterial effect. It has been reported to inhibit staphylococci, streptococci, and *Enterobacteriaceae*, although with more effectiveness against Gram-positive organisms (25, 58–61), probably due to the lack of a protective outer membrane barrier (62). Its antibacterial effect requires its incorporation into RNA at transcription, thereby interfering with thymine synthesis (59), but the inhibition of cell wall synthesis was also reported (63). Long-term *in vitro* exposure to 5-fluorouracil did not change the antibiotic susceptibility in *P. aeruginosa* strains (64). In addition, it could be a good candidate for *in vivo* use as an antivirulence drug against *P. aeruginosa*. Indeed, exposure to 5-fluorouracil inhibited the production of several QS-controlled virulence factors in *P. aeruginosa* clinical strains, including the production of biofilm elastase, rhamnolipids, pyoverdine, pyocyanin, swarming motility, protease, and exotoxin A (35, 65, 66). Herein, we observed that 5-fluorouracil was the most active hit in preventing biofilm formation by *P. aeruginosa*, regardless of the concentration tested.

It is not surprising that most of the candidate hits were anticancer drugs. Indeed, growing tumors and bacterial infections share several properties, such as the marked tendency to spread, replicate, and develop resistance to the immune system and chemotherapeutic agents (67). In addition, cancer cells might use cell-cell communication systems, comparable to the QS of bacterial cells, to coordinate their attacks against the host successfully (68). Repositioning of anticancer drugs for infectious diseases has been reported (69). Notably, a phase II study was started in 2016 to assess intravenous (i.v.) gallium nitrate's efficacy in improving pulmonary function in adult CF patients with *P. aeruginosa* infection (https://clinicaltrials.gov/ct2/show/NCT02354859). However, cytotoxicity and side effects associated with most anticancer drugs are reasonable concerns for using these drugs to treat infectious diseases. Therefore, treatment must balance the therapeutic benefit with potential side effects before considering the use of anticancer drugs for severe infectious diseases. Due to the pressing need for new molecules to use as a "last resort" to treat lethal MDR bacterial infections, such as those established in CF lung, anticancer agents might be considered in terminal cases where their beneficial effect would outweigh any potential side effect.

**Selected repositionable candidate hits: comparison with pharmacokinetic data from the literature.** To better understand the clinical potential of our hits, we compared our results with pharmacokinetic data from literature studies (Fig. 11).

The $MIC_{50}$ value observed for tirapazamine (8.9 $\mu$g/mL) was higher than achievable concentrations in humans. Indeed, earlier pharmacokinetics data on tirapazamine revealed a maximum concentration of 5.9 $\mu$g/mL after continuous intravenous infusion for 2 h of 260 mg/mm$^2$ (70). Similarly, $MIC_{50}$ and $MIC_{90}$ values of tavaborole, carmofur, and 5-fluorouracil were much higher than the plasma concentrations found in humans following oral or intravenous routes (61–74). However, pulmonary delivery might overcome these drawbacks, especially in the cases of 5-fluorouracil and tavaborole, due to the high therapeutic indexes we observed. It might allow the delivery of a minimum effective drug concentration directly to the lungs, thereby minimizing adverse effects and the potential selection of drug-resistant strains.

In this framework, various inhaled antibiotics have been approved to treat chronic *P. aeruginosa* lung infection in CF patients (9, 75). Particularly, CF treatment guidelines recommend the use of inhaled tobramycin to reduce exacerbations and to improve lung function and quality of life in CF patients (age of ≥6 years) with moderate-to-severe lung disease with chronic *P. aeruginosa* infection (9). No data are available for

| Hits | Structural and chemical formula | MIC$_{50}$ and MIC$_{90}$ (µg/mL) | MNCC$^a$ (µg/mL) | Anti-biofilm activity (range, µg/mL) | Data from human studies [reference] |
|---|---|---|---|---|---|
| Ebselen | $C_{13}H_9NOSe$ | MIC$_{50}$: 27.4 <br> MIC$_{90}$ > 54.8 | 54.8 | Biofilm formation: 3.4-6.8 | Still in clinical trials. Pharmacology and pharmacodynamics studies are in progress |
| Tirapazamine | $C_7H_6N_4O_2$ | MIC$_{50}$: 8.9 <br> MIC$_{90}$: 35.6 | 17.8 | Biofilm formation: 2.2-4.4 <br> Preformed biofilm: 35.6 | $C_{max}$ 5.97 µg/mL by a continuous i.v. infusion for 2 h of 260 mg/m$^2$ [70] |
| Tavaborole | $C_7H_6BFO_2$ | MIC$_{50}$: 15.1 <br> MIC$_{90}$ > 30.2 | 120.8 | Biofilm formation: 1.8-3.7 <br> Preformed biofilm: 30.2 | $C_{max}$ 5.9 ng/mL by a single topical application of a 5% (*v/v*) solution [74] |
| Carmofur | $C_{11}H_{16}FN_3O_3$ | MIC$_{50}$: 25.7 <br> MIC$_{90}$: 51.5 | 51.4 | Biofilm formation: 3.2-6.4 | Serum concentration < 0.02 µg/mL by an oral daily dose of 600 mg for more than 28 days [71] |
| 5-Fluorouracil | $C_4H_3FN_2O_2$ | MIC$_{50}$: 3.25 <br> MIC$_{90}$: 26 | 52 | Biofilm formation: 1.6-3.2 | $C_{max}$ 0.39–0.49 µg/mL with an oral dose of 50 mg/kg [67]. Serum concentration 0.07-0.39 µg/mL by a continuous i.v. infusion for 24 h [73] |

**FIG 11** Selected "most-promising" candidate hits revealed in the present study. MIC$_{50}$, MIC$_{90}$, cytotoxicity, and biofilm data were from the present study; *in vivo* pharmacokinetic data were from the literature. MNCC$^a$, maximum noncytotoxic concentration.

ebselen since it is currently in clinical trials; therefore, pharmacology and pharmacodynamics studies are still in progress.

**Conclusions.** From a drug repurposing point of view, identification of drugs with no known antibacterial activity that demonstrated excellent efficacy against *P. aeruginosa* opens a valuable new avenue for the rapid development of specific antibacterial agents, which are urgently needed for the management of CF patients.

To ensure the best translational applicability for defining novel treatments, models that more accurately represent *in vivo* conditions are required. In this framework, our small-scale CF-oriented assay provides a simple high-throughput platform for generating meaningful data to orient drug development adequately. Specifically, the most promising molecules that appeared from the HTS—performed for the first time under "CF-like" experimental conditions—were ebselen, tirapazamine, carmofur, tavaborole, and 5-fluorouracil. They displayed relevant antibacterial and antibiofilm activity toward MDR and virulent *P. aeruginosa* at concentrations not toxic for CF bronchial cells. These findings highlight their potential in treating *P. aeruginosa* lung infection in CF patients. In addition to the antibacterial activity, when viewed in the context of available literature, these molecules show other properties—e.g., antivirulence and anti-inflammatory—that could be relevant for managing CF patients.

Although these hits will require further investigation, validation, and extensive animal studies to examine their efficacy and toxicity before potential clinical use, these compounds may supply information for new targets and better insight into bacterial defense mechanisms for development of new antibiotics. In addition, drug combination and chemical modification might allow further exploration to improve their antibacterial activity and toxicity.

## MATERIALS AND METHODS

**Library and chemicals.** The Drug Repurposing Compound Library (catalog no. HY-L035) was purchased from MedChem Express (Monmouth Junction, NJ, USA). The library was provided in a 96-well plate format with aliquots of 10 mM stocks of drugs in dimethyl sulfoxide (DMSO) or water, stored at −80°C. Active compounds at HTS antibacterial assay—so-called "hits"—were purchased, as a powder of known potency, from MedChem Express. Tobramycin was from Sigma-Aldrich and was used as the

reference antibiotic. Stock solutions of 10 mM were prepared for each hit according to the manufacturer's instructions, aliquoted, and stored at −80°C until use.

**Bacterial strains and growth conditions.** The *P. aeruginosa* RP73 strain was used for the initial HTS. Isolated 16.9 years after the onset of infection in a CF patient from the Hannover cohort (76), this MDR nonmucoid strain is widely used as a prototype for studying chronic illness. The antibacterial activity of the hit compounds was evaluated more in depth using an additional set of 18 strains: *P. aeruginosa* BJ 3525, MG 3871, FM 3986, GR 3957, RI 3988, CA 3983, SM 3483, BA 3671, NF 3975, and VA 3949; *S. aureus* Sa2 and Sa16; *S. maltophilia* SanG and Sm142; *B. cepacia* Bc11 and Bc23; and *A. baumannii* Ab63 and Ab219. All strains were isolated from CF patients at the Fondazione IRCCS Ca' Granda Ospedale Maggiore Policlinico, Microbiology Unit, Milan, Italy. Notably, *P. aeruginosa* strains were selected as the representative for different infection statuses (i.e., first, sporadic, and chronic), antibiotic resistance levels and virulence phenotypic traits (i.e., biofilm formation; pyocyanin, pyoverdine, and protease production) (see Table S1 in the supplemental material). A completed list of genotypic features obtained by whole-genome sequencing analysis is also reported in Table S2.

Some colonies grown on tryptone soya agar (TSA) (Oxoid, Milan, Italy) following overnight incubation at 37°C were suspended in 0.9% sterile saline (Fresenius Kabi Italia, Verona, Italy) to reach an optical density at 500 nm ($OD_{500}$) of 0.150. This suspension was diluted 1:10 in sterile saline to achieve a final concentration of $1 \times 10^7$ to $2 \times 10^7$ CFU/mL. This standardized inoculum was used for all the assays unless indicated differently.

**"CF-like" experimental conditions.** To simulate the physicochemical properties seen in the CF airways, all the assays were carried out under "CF-like" conditions, namely, in an artificial sputum medium (ASM) (15), under acid conditions (pH 6.8) (77), and under a 5% $CO_2$ atmosphere. ASM had a composition that closely resembles CF sputum, according to Sriramulu et al. (15), with some modifications (all ingredients were from Sigma-Aldrich, unless otherwise specified): 5 g mucin 2 from pig stomach type, 4 g DNA from herring sperm, 5.9 mg diethylene triamine pentaacetic acid, 5 g NaCl, 2.2 g KCl, 0.1 g Tris-HCl, 5 mL egg yolk emulsion, and 5 g Casamino Acids (Difco) per L of water.

**Antibacterial HTS assay.** The library was screened at a single concentration point to identify antibacterial hit compounds against the *P. aeruginosa* RP73 strain. Briefly, 5 $\mu$L of the standardized inoculum (corresponding to $0.5 \times 10^5$ to $1 \times 10^5$ CFU/well) were added to each well of a 96-well polystyrene microtiter plate containing 94 $\mu$L of ASM with 1 $\mu$L of a 10 mM compound stock solution from the MedChem library, achieving a final drug concentration of 0.1 mM. Uninoculated samples with 1% (vol/vol) DMSO (final background in each well) were considered blank. Negative control was also prepared with 50% (vol/vol) DMSO to yield 100% killing. The content of each well was mixed by pipetting, and plates were incubated at 37°C under 5% $CO_2$. After 24 h-incubation, the survival rate of planktonic cells was assessed spectrophotometrically by two methods using a Tecan Sunrise microplate reader (Tecan Group Ltd. Mannedorf, Switzerland). First, the broth culture phase's optical density at 620 nm ($OD_{620}$) was measured. Afterward, each well was added with 20 $\mu$L of CellTiter AQueous One solution assay (Promega Italia, Milan, Italy), and after 1.5 h of incubation in the dark at 37°C, the $OD_{492}$ was measured. This value was corrected by subtracting the average $OD_{492}$ value of the uninoculated wells (blank). The percentage of surviving cells was calculated compared to the inoculated, but not treated, control sample (100% survival). The antibacterial activity of library drugs was classified based on the growth reduction compared to the untreated control sample: (i) low efficacy, <25%; (ii) moderate efficacy, $25\% \leq x < 60\%$; (iii) high efficacy, $60\% \leq x < 90\%$; and (iv) excellent efficacy, $90\% \leq x \leq 100\%$. Only drugs causing a ≥90% reduction of bacterial burden were considered potential anti-*P. aeruginosa* hit compounds.

**HTS assay validation.** The results from each HTS microplate were validated by calculating the *Z*-factor. To validate the degree of separation, the *Z*-factor and the percentage of inhibition of the positive and negative controls were determined using the formula:

$$Z\text{-factor} = 1 - \frac{3(\sigma_p + \sigma_n)}{\mu_p - \mu_n}$$

where $\sigma_p$ and $\sigma_n$ are the standard deviations of the positive and negative controls, respectively, and $\mu_p$ and $\mu_n$ are the corresponding mean values. A *Z*-factor between 0.5 and 1.0 indicates an excellent assay and statistically reliable separation between the positive and negative controls.

**MIC and MBC assays.** The MICs of the antibacterial hits against several CF strains of *P. aeruginosa*, *S. aureus*, *A. baumannii*, *S. maltophilia*, and *B. cepacia* were measured, comparatively to tobramycin, by a standard broth microdilution assay, according to the CLSI guidelines (78). *P. aeruginosa* ATCC 27853 and *S. aureus* ATCC 25923 were used as quality control strains. To determine MBCs, 10 $\mu$L from each well showing no growth at MIC reading was mixed gently and streaked on TSA, and then the plates were incubated at 37°C for colony count. The MBC value was the minimum drug concentration causing ≥99.9% killing on the plate after 24 h of incubation. Differences in MIC or MBC values were considered statistically significant if they were ≥2 $\log_2$.

**Cytotoxicity assay.** IB3-1 bronchial epithelial cells (ATCC CRL-2777) were grown as a monolayer at 37°C, under a 5% $CO_2$ atmosphere, in LHC-8 medium (Thermo Fisher Scientific Italia, Milan, Italy) supplemented with 5% (vol/vol) fetal bovine serum (Gibco, Milan, Italy). The cells were exposed for 24 h to different concentrations of each hit, while untreated IB3-1 cells were used as control. After incubation, the cells were washed with prewarmed phosphate-buffered saline (PBS) (Oxoid) three times, and viability was measured by an MTS-based colorimetric assay (CellTiter 96 AQueous One solution cell proliferation assay; Promega, Milan, Italy). After 4 h of incubation at 37°C under an atmosphere with 5% $CO_2$, the absorbance of each well was measured at 490 nm using a Tecan Sunrise microplate reader. Data were normalized using the positive control absorbance value as 100% viability and the absorbance value of 5%

DMSO as 0% viability. The concentration required to kill cells by 50%, compared to that of untreated control (growth medium only), was then calculated by nonlinear regression (variable slope) using GraphPad Prism 7.0 software (GraphPad Software, Inc., San Diego, CA, USA) and reported as the $LC_{50}$.

**Time-kill curve assay.** The standardized inoculum of *P. aeruginosa* RP73 was exposed to each hit compound tested at the MIC value and its multiples. During incubation at 37°C under shaking conditions, aliquots were taken from the samples at periodic intervals (2, 4, 6, and 24 h), serially diluted in sterile saline, and plated on TSA. After incubation for 24 h at 37°C, cell viability was determined by CFU counting. A drug was considered bactericidal if it reduced the initial inoculum by 3 log CFU or more within 24 h.

**Checkerboard assay.** The standardized inoculum was distributed into 96-well microplates, and a two-dimensional chessboard was established by adding serial dilutions of tobramycin and each hit horizontally and vertically, with a final volume of 100 $\mu$L per well. Each compound was tested between 1/128× and 2× MIC. After incubation at 37°C for 20 h, the results were determined using CellTiter 96 AQueous One solution cell proliferation assay, as described above. The fractional inhibitory concentration index (FICI) was calculated as [(MIC of drug A in combination/MIC of drug A alone)] + [(MIC of drug B in combination/MIC of drug B alone)] and interpreted as follows: FICI $\leq$ 0.5 = synergy, 0.5 < FICI $\leq$ 1.0 = additivity, 1.0 < FICI < 4.0 = indifference, and FICI $\geq$ 4.0 = antagonism (79).

**Biofilm inhibition and eradication assays.** A broth culture prepared in tryptone soya broth (TSB) (Oxoid) (37°C, 130 rpm, 16 h) was corrected to an $OD_{550}$ of 0.8 (0.5 × $10^9$ to 1.0 × $10^9$ CFU/mL) and diluted 1:100 with ASM. To evaluate the effect of the drugs in preventing biofilm formation, 200 $\mu$L of this suspension was added in each well of a TC-treated microplate (Falcon) with (1/2× and 1/4× MIC) or without (control) the desired drug. After incubation at 37°C for 24 h, planktonic cells were gently removed by washing twice with 200 $\mu$L PBS. After fixing samples (60°C, 1 h), the biofilm biomass was quantified with crystal violet staining. Briefly, 200 $\mu$L of 10% (wt/vol) Hucker's crystal violet was added to each well, and after incubation at room temperature for 5 min, each well was washed using tap water. After drying at 37°C, 200 $\mu$L 33% (vol/vol) glacial acetic acid was added to dissolve the stained dye for 15 min. The biofilm biomass was determined by measuring the absorbance at 492 nm using a Tecan Sunrise microplate reader. To evaluate the efficacy of drugs against preformed biofilms, 24-h biofilms were treated with 200 $\mu$L of ASM with (2× and 4× MIC) or without (control) the selected drug at 37°C for another 24 h and then washed with PBS. As described above, crystal violet staining finally measured the biofilm biomass.

**Microscopic analysis.** The effects of the selected hits on *P. aeruginosa* RP73 morphology were assessed by FIB/SEM. Cells were cultured overnight in the presence or the absence of each hit at 2× MIC, harvested by centrifugation (10,000 × $g$, 5 min), washed twice with PBS, and fixed with 2.5% glutaraldehyde (vol/vol) in 0.1 M cacodylate buffer (pH 7.4) for 1 h at 4°C. Cells were then recovered by centrifugation (10,000 × $g$, 5 min) and washed in cacodylate buffer (3 times, 10,000 × $g$, 4°C), and the pellet was postfixed with 1% (vol/vol) osmium tetroxide in 0.2 M cacodylate buffer for 1 h at room temperature. The samples were dehydrated with graded ethanol solutions (vol/vol) (70% for 10 min, 80% for 10 min, 95% for 10 min, and twice in 100% for 15 min), *en bloc* stained with 2% (vol/vol) ethanolic uranyl acetate, embedded in Epon 812 resin, and left to polymerize for 3 days at 60°C. Thick sections of roughly 15 $\mu$m were secured to aluminum stubs using double-sided carbon disks, gold sputter coated, and observed by a Fei Helios Nanolab 600 at the Interdepartmental Laboratory of Electron Microscopy (LIME) (Roma Tre University, Rome, Italy).

**Determination of cell membrane integrity.** Briefly, $10^8$ cells grown in TSB up to log phase were washed twice (4,200 rpm, 10 min, 24°C) with 0.9% NaCl, resuspended in 0.9% NaCl, and treated with each compound at serial concentrations (1×, 2×, 4×, and 8× MIC). Cells exposed to 70% (vol/vol) isopropyl alcohol (Sigma-Aldrich) or NaCl 0.9% were used as positive and negative controls, respectively. After 2 h of incubation at 37°C, the bacterial suspension was washed twice with 0.9% NaCl and aliquoted (100 $\mu$L/well) in a flat-bottomed, 96-well, black microtiter plate (Corning, Turin, Italy). Cells were labeled with 100 $\mu$L PI (Thermo Fisher Scientific Italia) to a final concentration of 30 $\mu$M and incubated in the dark at room temperature for 15 min. The fluorescence intensity was measured ($\lambda$ex = 485 nm, $\lambda$em = 630 nm) using a fluorescence microplate reader (Tecan Infinite M Plex).

**Membrane depolarization assay.** In brief, $10^8$ cells grown in TSB up to log phase were washed twice (4,200 rpm, 10 min, 24°C) with 5 mM HEPES (pH 7.2) (Sigma-Aldrich) plus 20 mM glucose (Sigma-Aldrich). Cells were suspended to an $OD_{550}$ of 0.06 in the same washing buffer added with 0.1 M KCl. Each well of a flat-bottomed, 96-well, black microtiter plate (Corning) had added to it 100 $\mu$L of the bacterial suspension and the membrane potential-sensitive dye DiSC3(5) (Sigma-Aldrich) at a final concentration of 2 $\mu$M. The microplate was incubated at room temperature in the dark for 1.5 h to favor the incorporation of the dye into the bacterial membrane. Each compound was added (5 $\mu$L/well) at serial concentrations (1×, 2×, 4×, and 8× MIC). Polymyxin B (Sigma-Aldrich) at 320 $\mu$g/mL and HEPES plus 0.1% DMSO were used as the positive and negative controls, respectively. The fluorescence intensity was measured ($\lambda$ex = 622 nm, $\lambda$em = 670 nm) using a fluorescence microplate reader (Tecan Infinite M PLEX).

**Total ROS measurement.** Briefly, $10^8$ cells grown in TSB up to log phase were washed once (4,200 rpm, 10 min, 24°C) with PBS and resuspended in the same buffer to an $OD_{550}$ of 0.75 (1 × $10^8$ to 5 × $10^8$ CFU/mL). This suspension was diluted 1:100 with PBS and incubated at 37°C with 100 $\mu$M 2′,7′-dichlorofluorescein diacetate (H2DCFHDA) for 2 h. After washing once with PBS, 190 $\mu$L of bacterial cells labeled with the probe and 10 $\mu$L of each compound tested at serial concentrations (1×, 2×, 4×, and 8× MIC) were added to each well of a flat-bottomed, 96-well, black microtiter plate (Corning). Hydrogen peroxide (100 $\mu$M) (Sigma-Aldrich) and *N*-acetylcysteine (5 mM) (Sigma-Aldrich) were used as the positive and negative controls, respectively. The fluorescence intensity was measured ($\lambda$ex = 488 nm, $\lambda$em = 525 nm) using a fluorescence microplate reader (Tecan Infinite M Plex).

**Statistical analysis.** All assays were conducted in triplicate and repeated twice ($n = 6$). Statistical analysis was performed using GraphPad Prism 7.0 software (GraphPad Software, San Diego, CA, USA). The data were normally distributed, as from the Shapiro-Wilk test. Ordinary one-way analysis of variance (ANOVA) with Holm-Sidak's (toxicity assays), Dunnett's (biofilm assays, mechanism of action assays), or Tukey's (mechanism of action assays) multiple-comparison test was applied when assessing differences between 3 or more groups of unpaired data. The statistical analysis assumed a confidence level of ≥95%, thus considering $P$ values of $<0.05$ statistically significant.

## SUPPLEMENTAL MATERIAL

Supplemental material is available online only.
**SUPPLEMENTAL FILE 1**, DOCX file, 0.1 MB.

## ACKNOWLEDGMENTS

This study was partly supported by the G. d'Annunzio University of Chieti-Pescara (FAR, 2021). The funders had no role in study design, data collection and interpretation, or the decision to submit the work for publication.

We warmly thank Sherry Lee Jones (RN case manager, Blue Cross Blue Shield, Michigan, USA) for reviewing the manuscript for English usage.

A. Pompilio and G. Di Bonaventura designed this study. L. Cariani collected the samples and isolated *P. aeruginosa* strains. A. Pompilio, G. Di Bonaventura, and V. Lupetti performed HTS, time-kill, antibacterial, checkerboard, cytotoxic, antibacterial mechanism, and antibiofilm assays. A. Di Giulio and M. Muzzi carried out microscopic analysis. A. Piccirilli performed bacterial genotyping. A. Pompilio and G. Di Bonaventura analyzed and interpreted the data. A. Pompilio and G. Di Bonaventura wrote the manuscript. V. Lupetti, A. Piccirilli, A. Di Giulio, and M. Muzzi revised the manuscript. All authors of this study have agreed to, read, and approved the manuscript and have given their written consent for submission and subsequent publication of the manuscript.

We declare no conflict of interest.

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
