## [Reviewer comments · Microbiology Spectrum]

Microbiology Spectrum

Repurposing high-throughput screening identifies unconventional drugs with antibacterial and antibiofilm activities against *Pseudomonas aeruginosa* under experimental conditions relevant to cystic fibrosis.

Giovanni Di Bonaventura, Veronica Lupetti, Andrea Di Giulio, Maurizio Muzzi, Alessandra Piccirilli, Lisa Cariani, and Arianna Pompilio

Corresponding Author(s): Giovanni Di Bonaventura, University of Chieti-Pescara

Review Timeline:

Submission Date:	January 22, 2023
Editorial Decision:	February 22, 2023
Revision Received:	April 28, 2023
Accepted:	May 13, 2023

Editor: Silvia Cardona

Reviewer(s): Disclosure of reviewer identity is with reference to reviewer comments included in decision letter(s). The following individuals involved in review of your submission have agreed to reveal their identity: Marina Rautenbach (Reviewer #2)

Transaction Report:

DOI: <https://doi.org/10.1128/spectrum.00352-23>

February 22, 2023

Prof. Giovanni Di Bonaventura
University of Chieti-Pescara
Chieti
Italy

Re: Spectrum00352-23 (Repurposing high-throughput screening identifies unconventional drugs with antibacterial and antibiofilm activities against *Pseudomonas aeruginosa* under experimental conditions relevant to cystic fibrosis)

Dear Prof. Giovanni Di Bonaventura:

Thank you for submitting your manuscript to Microbiology Spectrum. Your article was evaluated by two experts in the field. Both reviewers found merit in your work. However, reviewers have raised concerns that need to be addressed before the article can be considered for publication. One reviewer suggested that high-throughput screening using clinically-relevant conditions is interesting but the work should show whether the hits could have also been found in standard conditions. I agree with this reviewer and I encourage you to discuss this point thoroughly. In addition, please ensure you address the limitations of the compounds found as their therapeutic index is modest.

Link Not Available

Sincerely,

Silvia Cardona

Journals Department
Reviewer comments:

Reviewer #1 (Comments for the Author):

The manuscript by Di Bonaventura et al. screened a library of previously approved drugs and other molecules currently at

various stages of the drug development pipeline for antimicrobial activity against *Pseudomonas aeruginosa*. *P. aeruginosa* is the most predominant pathogen affecting patients with cystic fibrosis (CF) and the authors used a previously developed CF sputum-like medium for the screen. Potential hits were evaluated for anti-biofilm activity and toxicity against CF bronchial cells. TEM studies evaluated the effects of the compounds against the bacterial membranes as a possible mechanism of their action. While the study did not reveal significantly more potent non-toxic hits (relative to the control, tobramycin), it may serve as a proof of concept for screening in clinically-relevant conditions. However, several limitations in the study have been identified as follows:

Major comments:

I. The authors did not demonstrate the benefit of screening under CF-like conditions (as opposed to the standard AST conditions, which is presumably easier). Would the hits in this study be discovered if the screen was conducted under standard conditions? The activity of the hits should be evaluated in MHB or another standard medium. E.g., it might be expected that Tirapazamine, as a prodrug (L. 386) would have no or low potency under standard media and is detected in CF-sputum due to being relatively hypoxic. This should be tested and discussed.

II. The therapeutic index seems to be marginal. The therapeutic index (L. 214) should also be calculated using the MIC90 and then determine if any of the five hits still satisfy the statement at the end of this section. More importantly, tobramycin's toxicity was not attained at any of the tested concentrations. Higher concentrations should be tested to determine the exact MNCC and hence its more accurate therapeutic index as a control antibiotic that is used both systemically and, in an aerosol/nebulized form. These calculations will further enable the assessment of the potential of these hits.

III. The mechanistic studies are very preliminary. The TEM shows some evidence of possible membrane permeabilization, other more quantitative methods should be used to quantify the leakage of cytoplasmic effects and the membrane damage. Membrane integrity assays (for e.g., fluorescent dye-based assays like NPN) should be carried out in dose-response assays to evaluate how the membrane integrity effects correlate with the observed antibacterial effects. In addition, other potential mechanisms should be explored. Some examples for potential hypotheses that could be followed up on are already mentioned/cited in the manuscript. For examples, see: L. 363 (Ebselen: ROS-mediated effects in other organisms could be tested against Pa); L. 379-380 for Tirapazamine; L. 403-404 for Tavaborole; and L. 413 for 5-fluorouracil.

Minor comments:

1. Line 101: Tobramycin wasn't used in the screen. So it is not clear what was tobramycin selected for in this section.
2. L 184-185: The results supporting this statement should be shown (antivirulence activity). Otherwise, the statement should be removed.
3. L. 188: "less changeable" not clear what does that mean.
4. L. 235: cfu counts of tavaborole increased at all tested concentrations (Fig 5). The statement should be revised accordingly.
5. L. 289-292: sentence structure needs to be revised.
6. L. 293: "least" not "last"
7. L. 334-338: sentence structure needs to be revised.
8. L. 393: Could you comment on the potential reason of the difference between the previously-reported lack activity against *P. aeruginosa* and the activity observed here? For example, how was carmofur's activity evaluated before? i.e., in what conditions, CF-like or MHB?
9. L. 434-436: sentence structure needs to be revised.
10. L. 446: MIC50 for Tirapazamine appears to be higher than achievable in humans after a rather extensive IV regimen. Thus, the sentence should be revised.
11. Fig 1: the legend should explain the difference between the red and blue labeling of hits.
12. Fig 2 and 3: color scale of the heat map could be more accessible.
13. Fig 6 and 7: Please show the data points in the bar graphs. Also, for Fig 7, could you speculate on why the effects are mostly less pronounced or absent at 4x MIC compared to at 2x?

Reviewer #2 (Comments for the Author):

This manuscript describes the search for anti-*Pseudomonas aeruginosa* compounds in a library of drug library consisting of 3386 drugs using a high throughput screening method tailored to conditions experience in cystic fibrosis (CF) The manuscript generally reads well, but there is a need for polishing the scientific English writing. The experimental approach is highly commendable, and the research results are very interesting with excellent potential to contribute to repurposing one of more of the hits from this study in the treatment of *P. aeruginosa* in CF patients. I have a number of queries and suggestions outlined below to improve the manuscript.

Queries

1. Fig. 4: Present data as semi-log dose response with sigmoidal fit. From this dose response inhibition parameters such as the 50% lethal concentration (LC) and LC can be calculated.
2. L233: What is "an almost bactericidal" activity? A antibacterial compound is bactericidal, bacteriostatic or inactive.
3. L262: The statement "regardless the concentration" is confusing. I assume that the compound is active over the whole concentration range tested. Rephrase.
4. Fig. 8: Why is travaborole less active against biofilms at 4xMIC than 2x MIC? Could it be a solubility issue at higher concentrations? Please explain.
5. What is the concentration unit(s) in Table 2? Supply activity parameters in mM to ease comparison of compound activity in Table 4.
6. Methodology must not be described in such detail in figure legends
7. There is a problem throughout the manuscript the use of use of capital letters for compound/drug names. Drug and reagent names in sentences must be in lower case. Check manuscript.
8. Care must be taken in the description of concentrations, specifically define *m/v*, *v/v* or *m/m* (must be in italic font) when expressing a concentration as a percentage.
9. Check for consistency in SI unit abbreviations, for example the SI-unit abbreviation for litre is L (capital L) (check mL and μ L).
10. Re-evaluate the use of significant numbers in methodology and data throughout the thesis. When referring to biological data and methods, the use of significant numbers in data must be re-evaluated in relation to the accuracy of the methodology used. If pipettes, analytical scales, spectrophotometers, detectors and imagers are used, the accuracy of data normally deteriorate to two significant digits (ie 0.23 or 2.3 or 230) and can seldom exceed three significant digits (ie 0.234 or 2.34 or 234).
11. There are some English language/scientific writing issues. Check whole manuscript for grammar and scientific writing issues.

Staff Comments:

Preparing Revision Guidelines

Please return the manuscript within 60 days; if you cannot complete the modification within this time period, please contact me. If you do not wish to modify the manuscript and prefer to submit it to another journal, please notify me of your decision immediately so that the manuscript may be formally withdrawn from consideration by Microbiology Spectrum.

Dear Reviewers,

Thank you for the review and comments and for allowing us to improve our manuscript to *Microbiology Spectrum*. We appreciate the time and effort that you have dedicated to providing your valuable feedback on our manuscript. We have been able to incorporate changes to reflect all the suggestions you provided. We have highlighted the changes within the manuscript. Here are responses to comments and concerns (line numbers are referred to the "Marked up MS – for review only" file).

REVIEWER #1 (Comments for the Author):

The manuscript by Di Bonaventura et al. screened a library of previously approved drugs and other molecules currently at various stages of the drug development pipeline for antimicrobial activity against *Pseudomonas aeruginosa*. *P. aeruginosa* is the most predominant pathogen affecting patients with cystic fibrosis (CF) and the authors used a previously developed CF sputum-like medium for the screen. Potential hits were evaluated for anti-biofilm activity and toxicity against CF bronchial cells. TEM studies evaluated the effects of the compounds against the bacterial membranes as a possible mechanism of their action. While the study did not reveal significantly more potent non-toxic hits (relative to the control, tobramycin), it may serve as a proof of concept for screening in clinically-relevant conditions. However, several limitations in the study have been identified as follows:

Major comments:

I. The authors did not demonstrate the benefit of screening under CF-like conditions (as opposed to the standard AST conditions, which is presumably easier). Would the hits in this study be discovered if the screen was conducted under standard conditions? The activity of the hits should be evaluated in MHB or another standard medium. E.g., it might be expected that Tirapazamine, as a prodrug (L. 386) would have no or low potency under standard media and is detected in CF-sputum due to being relatively hypoxic. This should be tested and discussed.

Response: We thank the Reviewer for pointing out this matter. In the revised MS, the antibacterial activity of the selected 14 hits has also been tested in "standard" conditions – i.e., cation-adjusted Muller-Hinton broth, neutral pH, and aerobic atmosphere. Our findings, summarized in the new Figure 2, clearly indicated that experimental conditions used significantly affect antibacterial activity, thus highlighting the need for using physico-chemical conditions relevant to the infection site. A new paragraph ("Impact of the experimental conditions on the hits' activity") has been added in the "Results and Discussion" section of the revised MS (Lines 173-191), along with 3 new references (#18, #19, #20).

II. The therapeutic index seems to be marginal. The therapeutic index (L. 214) should also be calculated using the MIC₉₀ and then determine if any of the five hits still satisfy the statement at the end of this section. More importantly, tobramycin's toxicity was not attained at any of the tested concentrations. Higher concentrations should be tested to determine the exact MNCC and hence its more accurate therapeutic index as a control antibiotic that is used both systemically and, in an aerosol/nebulized form. These calculations will further enable the assessment of the potential of these hits.

Response: We thank the Reviewer for the comment. The therapeutic index has also been calculated on the MIC₉₀ values and results have been summarized in the updated Table 2. In addition, as suggested by the Reviewer, tobramycin's toxicity towards IB3-1 bronchial cells has also been tested over an extended range of concentrations, to evaluate its accurate therapeutic index; results are shown in Figures 5 and S1, and in Table 2. As hypothesized by the Reviewer, after calculating MNCC/MIC₉₀ values (Lines 250-255), the therapeutic index seems to be "marginal". We referred at this matter in the "Results and Discussion" section, although we think it is worth highlighting the advantageous MNCC/MIC₅₀ index associated to 5-fluorouracil (Lines 256-261).

III. The mechanistic studies are very preliminary. The TEM shows some evidence of possible membrane permeabilization, other more quantitative methods should be used to quantify the leakage of cytoplasmic effects and the membrane damage. Membrane integrity assays (for e.g., fluorescent dye-based assays like NPN) should be carried out in dose-response assays to evaluate how the membrane integrity effects correlate with the observed antibacterial effects. In addition, other potential mechanisms should be explored. Some examples for potential hypotheses that could be followed up on are already mentioned/cited in the manuscript. For examples, see: L. 363 (Ebselen: ROS-mediated effects in other organisms could be tested against Pa); L. 379-380 for Tirapazamine; L. 403-404 for Tavorole; and L. 413 for 5-fluorouracil.

Response: We thank the Reviewer for this valuable comment. Trying to gain new insights in the mechanism(s) underlying the antibacterial activity, the impact of exposure to the selected hits on cell membrane integrity and depolarization, and on ROS intracellular production has been evaluated by using quantitative methods (propidium iodide uptake, Disc3(5), and 2',7'-dichlorofluorescein diacetate assays), as suggested by the Reviewer. In the revised version of the MS, "Introduction" (Lines 95-96), "Materials and Methods" (Lines 733-763) and "Results and

Discussion" (Lines 374-435) sections have been implemented in accordance, along with the addition of a new Figure 9 summarizing all findings from the MoA-assays, and 2 new references (#39, #43).

Minor comments:

1. Line 101: Tobramycin wasn't used in the screen. So it is not clear what was tobramycin selected for in this section.

Response: The sentence has been moved to the end of the Introduction section (Lines 98-99).

2. L 184-185: The results supporting this statement should be shown (antivirulence activity). Otherwise, the statement should be removed.

Response: We wanted to state that 5-fluorouracil showed high activity against virulent strains rather than an antivirulence activity. Therefore, in the revised version the sentence has been rephrased for the sake of clarity, and a reference to Figure 3 has been added (Lines 210-211).

3. L. 188: "less changeable" not clear what does that mean.

Response: For greater clarity, it has been rephrased as "less variable" (Line 214).

4. L. 235: cfu counts of tavorole increased at all tested concentrations (Fig 5). The statement should be revised accordingly.

Response: The sentence has been changed based on the findings (Lines 277-279).

5. L. 289-292: sentence structure needs to be revised.

Response: The sentence has been rephrased for the sake of clarity (Lines 333-335). The specification that tirapazamine was tested only at 2 x MIC - because at 4 x MIC it was previously found to be toxic to IB3-1 cells – has been maintained in the legend of Figure 8.

6. L. 293: "least" not "last"

Response: It has been fixed (Line 339).

7. L. 334-338: sentence structure needs to be revised.

Response: For the sake of clarity, the sentence has been rephrased and shortened as follows: "A minority of bacteria showed the total loss of intracellular content, indented outer membrane, irregular/shrunken shape, and uneven or absent cytoplasmatic electron density." (Lines 383-384).

8. L. 393: Could you comment on the potential reason of the difference between the previously-reported lack activity against *P. aeruginosa* and the activity observed here? For example, how was carmofur's activity evaluated before? i.e., in what conditions, CF-like or MHB?

Response: The antibacterial activity of carmofur was observed by Peyclit et al. in cation-adjusted Muller-Hinton broth, not under "CF-like conditions". We addressed to this point at Lines 503-510.

9. L. 434-436: sentence structure needs to be revised.

Response: The sentence has been improved in the structure (Line 550-553).

10. L. 446: MIC50 for Tirapazamine appears to be higher than achievable in humans after a rather extensive IV regimen. Thus, the sentence should be revised.

Response: The sentence has been modified according to the Reviewer's comment (Lines 564-565).

11. Fig 1: the legend should explain the difference between the red and blue labeling of hits.

Response: The meaning of red and blue labeling was already explained in the Figure 1 legend of the original version, as follows: "The primary screening revealed 106 hits (red highlighted), 14 of which (blue highlighted) were identified as repositionable drug candidates". As a result, there were no changes to the legend in Figure 1.

12. Fig 2 and 3: color scale of the heat map could be more accessible.

Response: The color scale of the original Figures 2-3 (now Figures 3 and 4 in the revised MS) has been made more understandable by changing the colors used in the scale. In addition, MIC and MBC measurement unit (mM) were added.

13. Fig 6 and 7: Please show the data points in the bar graphs. Also, for Fig 7, could you speculate on why the effects are mostly less pronounced or absent at 4x MIC compared to at 2x?

Response: As required by the Reviewer, in the revised MS the data points were added in Figures 6-7.

Regarding the Figure 7 (re-numbered as Figure 8 in the revised MS):

- tavorole at 2xMIC has an antibiofilm effect less pronounced compared with that observed at 4xMIC. This could raise a solubility issue at higher concentrations. In addition, although tavorole is fully solubilized in DMSO-prepared stock solution, when diluted in ASM, the chemical complexity of the synthetic medium' recipe could raise a solubility issue. Unfortunately, the turbid nature of synthetic ASM makes it impossible to reveal the possibility of drug precipitation. In the absence of confirmation, we would prefer not to make any reference to this issue in the text.
- during MS revision, we realized that some statistically significant differences were lacking in Figure 7 (re-numbered as Figure 8 in the revised MS); therefore, in the revised MS, Figure 8 has been updated.

REVIEWER #2 (Comments for the Author):

This manuscript describes the search for anti-*Pseudomonas aeruginosa* compounds in a library of drug library consisting of 3386 drugs using a high throughput screening method tailored to conditions experience in cystic fibrosis (CF) The manuscript generally reads well, but there is a need for polishing the scientific English writing. The experimental approach is highly commendable, and the research results are very interesting with excellent potential to contribute to repurposing one of more of the hits from this study in the treatment of *P. aeruginosa* in CF patients. I have a number of queries and suggestions outlined below to improve the manuscript.

Queries

1. Fig. 4: Present data as semi-log dose response with sigmoidal fit. From this dose response inhibition parameters such as the 50% lethal concentration (LC) and LC can be calculated.

Response: In the revised manuscript (MS), Figure 4 of the original version (now moved in the "Supplemental material" as Fig. S1) has been replaced with a new one (Figure 5) summarizing the dose-effect curves and related LC₅₀ values. Methods were implemented accordingly (Lines 685-689), and LC₅₀ values mentioned in the "Results and Discussion" section (Lines 240-244).

2. L233: What is "an almost bactericidal" activity? A antibacterial compound is bactericidal, bacteriostatic or inactive.

Response: The sentence has been rephrased in accordance with the Reviewer's comments (Lines 276-279).

3. L262: The statement "regardless the concentration" is confusing. I assume that the compound is active over the whole concentration range tested. Rephrase.

Response: The sentence has been rephrased for the sake of clarity (Line 305-306).

4. Fig. 8: Why is tavorole less active against biofilms at 4xMIC than 2x MIC? Could it be a solubility issue at higher concentrations? Please explain.

Response: Decrease in tavorole anti-biofilm activity when tested at 4xMIC, compared with 2xMIC, might be due to a lower solubility at higher concentrations. In addition, although tavorole is fully solubilized in DMSO-prepared stock solution, when diluted in ASM, the chemical complexity of the synthetic medium' recipe could raise a solubility issue. Unfortunately, the turbid nature of synthetic ASM makes it impossible to reveal the possibility of drug precipitation. In the absence of confirmation, we would prefer not to make any reference in the text.

5. What is the concentration unit(s) in Table 2? Supply activity parameters in mM to ease comparison of compound activity in Table 4.

Response: The concentration unit in Table 2 was mM. In the revised MS, this information has been added to the Table's legend.

6. Methodology must not be described in such detail in figure legends

Response: The figure legends have been shortened by deleting the methodology information already cited in the text.

7. There is a problem throughout the manuscript the use of use of capital letters for compound/drug names. Drug and reagent names in sentences must be in lower case. Check manuscript.

Response: The drug and reagent names have been corrected to lowercase throughout the manuscript.

8. Care must be taken in the description of concentrations, specifically define *m/v*, *v/v* or *m/m* (must be in italic font) when expressing a concentration as a percentage.

Response: The manuscript has been carefully checked and improved according to the Reviewer' comment.

9. Check for consistency in SI unit abbreviations, for example the SI-unit abbreviation for litre is L (capital L) (check mL and μ L).

Response: All the abbreviations have been fixed according to the SI unit throughout MS.

10. Re-evaluate the use of significant numbers in methodology and data throughout the thesis. When referring to biological data and methods, the use of significant numbers in data must be re-evaluated in relation to the accuracy of the methodology used. If pipettes, analytical scales, spectrophotometers, detectors and imagers are used, the accuracy of data normally deteriorate to two significant digits (ie 0.23 or 2.3 or 230) and can seldom exceed three significant digits (ie 0.234 or 2.34 or 234).

Response: The text has been modified according to the Reviewer' comments.

11. There are some English language/scientific writing issues. Check whole manuscript for grammar and scientific writing issues.

Response: The manuscript has been reviewed for the English language by a native speaker we cited in the "Acknowledgements" section (Lines 788-789).

May 13, 2023

Prof. Giovanni Di Bonaventura
University of Chieti-Pescara
Chieti
Italy

Re: Spectrum00352-23R1 (Repurposing high-throughput screening identifies unconventional drugs with antibacterial and antibiofilm activities against *Pseudomonas aeruginosa* under experimental conditions relevant to cystic fibrosis.)

Dear Prof. Giovanni Di Bonaventura:

Your manuscript has been accepted, and I am forwarding it to the ASM Journals Department for publication. You will be notified when your proofs are ready to be viewed.

Please, note that I have indicated that you need to provide a Data Availability paragraph at the end of the Materials and Methods section with the name of the repository where the raw data can be accessed. I recommend submission of the results of the screening to Pubchem assay.

Sincerely,

Silvia Cardona
Editor, Microbiology Spectrum
